# CODE-ENABLED LANGUAGE MODELS CAN OUTPERFORM REASONING MODELS ON DIVERSE TASKS

## ABSTRACT

Reasoning models (RMs), language models (LMs) trained with reinforcement learning to produce long-form natural language reasoning, have been remarkably successful, but they still cost large amounts of compute and data to train and can be slow and expensive to run. In this paper, we show that ordinary LMs can already be elicited to be strong reasoners at a level comparable to or even surpass their corresponding RMs (e.g., DeepSeek V3 vs R1) without finetuning, across diverse domains from instruction following and creative generation to mathematical reasoning. This is achieved by combining the CodeAct approach, where LMs interleave natural language reasoning with code executions in a multi-step fashion, with few-shot bootstrap in-context learning—from as few as five training problems. Analyzing four matched pairs of LMs and RMs, we find that our framework, coined `CodeAdapt`, enables three LMs to outperform the corresponding RMs on average over eight tasks (up to 22.9%) while being 10-81% more token efficient, and delivers superior performance for six tasks on average over models (up to 35.7%). The code-augmented reasoning traces further display rich and varied problem-solving strategies. Our findings support that (1) CodeAdapt-style learning and reasoning may be domain general and robust and (2) code-enabled LMs are cognitively relevant and powerful systems, potentially providing a strong foundation for in-weight reinforcement learning.

## 1 INTRODUCTION

Language models (LMs) have rapidly become versatile general-purpose systems, yet their ability to reliably perform complex, multi-step reasoning remains a central challenge (Berglund et al., 2023; Wu et al., 2024; Phan et al., 2025). The advent of reasoning models (RMs) marks a significant milestone addressing this challenge: LMs trained via large-scale reinforcement learning (RL) to incentivize long-form natural language reasoning chains demonstrate remarkable gains on a wide range of reasoning domains (OpenAI et al., 2024; Guo et al., 2025; Gemini Team et al., 2025), and this paradigm works particularly well on tasks where completions are easily or objectively verifiable, such as math competitions and programming (Li et al., 2025; Chen et al., 2025b).

Nonetheless, the improvements come at substantial cost. Even though training DeepSeek R1, an exemplar reasoning model, from the base V3 model requires significantly less resources than training frontier non-reasoning models from scratch, the amount of data and compute is still prohibitive for most small organizations and academic entities (Guo et al., 2025). Furthermore, deploying RMs also escalates costs, as they can be slow and expensive to run—characteristic of their long reasoning chains and inference-time scaling behaviors (Sui et al., 2025), and because they are not superior to standard LMs on all tasks (Aggarwal et al., 2025), they are not one-size-fit-all to common use cases. These considerations raise a fundamental question: must we spend these resources to achieve advanced reasoning, or might there be more economical alternatives that reach comparable performance?

In this paper, we investigate whether a simple alternative approach that we call `CodeAdapt` can compete with expensive reasoning models: equipping standard LMs with iterative code execution capabilities and minimal in-context bootstrap learning. The approach combines CodeAct (Wang et al., 2024a)—which allows models to write and execute Python code across multiple conversational turns—with a lightweight training procedure using just five training problems per task domain. While CodeAct and similar agentic frameworks have primarily been developed and evaluated for

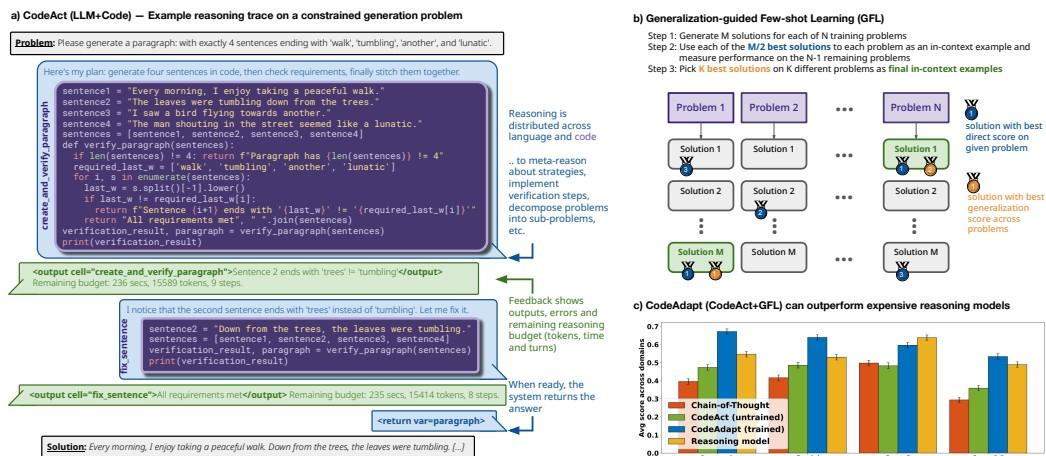

Figure 1: **CodeAdapt architecture and performance.** (a) We combine an iterative code-execution LM reasoning framework inspired by CodeAct (Wang et al., 2024a) with (b) a novel lightweight in-context learning procedure, and (c) show that this simple combination can help a set of LMs from different providers outperform corresponding reasoning models evaluated on eight diverse tasks.

action-driven tasks such as digital assistance and data analysis (Wang et al., 2025; Huang et al., 2025), and while prior work has noted code's potential for enhancing reasoning (e.g., Chen et al., 2023; Gao et al., 2023; Li et al., 2024), a critical question remains unexplored: can this iterated problem solving paradigm, combined with minimal domain adaptation, serve as a direct competitor to reasoning models, and if so on which tasks? Now the growing availability of open-weight reasoning models permits such comparisons, and we provide a systematic evaluation of this possibility: comparing four "instruct" LMs equipped with CodeAct and few-shot training against their reasoning-trained counterparts across eight diverse tasks spanning instruction following, language processing, and logic puzzles. Our results demonstrate that overall this combination consistently matches or outperforms reasoning models (up to 22.9% per model and 35.7% per task) without any finetuning or domain-specific scaffolding. CodeAdapt achieves this capability with a handful of training examples rather than substantial RL training overhead and 10-81% improved inference-time token efficiency.

We motivate our framework and interpret the finding through the lens of *extended cognition* (Clark & Chalmers, 1998): the idea that human cognitive processes are not confined to the boundaries of the brain but can extend into external tools and environments that become integral parts of the thinking process. In our setting, the code interpreter serves as a workspace where models can offload structured computations, verify hypotheses through execution, and build upon intermediate results—transforming code from a mere output format into an active reasoning substrate. This represents a form of *modular reasoning* (Mahowald et al., 2024), where models dynamically partition cognitive work: keeping high-level planning, intuitive reasoning and contextual understanding internal while externalizing precise calculations, control flow, iterative searches, and systematic verification to executable code. Rather than requiring models to internalize all reasoning patterns through large-scale training, this approach leverages the complementary strengths of neural language processing and symbolic computation, allowing models to reason *with* code rather than merely *about* code. It adapts to a domain efficiently without human demonstrations, consistent with the hypothesis that program-like representations explain how humans learn so fast (Lake et al., 2017; Rule et al., 2020).

Overall, our work makes three primary contributions. First, we present one of the first systematic empirical comparisons demonstrating that instruct LMs equipped with iterative code execution and minimal domain adaptation can match or exceed the performance of expensively trained reasoning models across diverse tasks. Second, we introduce CodeAdapt, which builds upon a lightweight but effective learning methodology that achieves this performance using only five training problems per domain, contrasting sharply with typical RL for reasoning. Third, we conduct in-depth examination of CodeAdapt, including ablation studies and analyses on reasoning patterns and resource usage—furthering our understanding of hybrid reasoning frameworks and their cognitive connections. More broadly, our work suggests that the path to better AI systems may lie not only in scaling compute and data (which CodeAdapt is synergistic with), but in designing richer cognitive architectures that effectively distribute learning and reasoning across multiple computational substrates.

## 2 RELATED WORK

**Reasoning with language models.** Our work contributes to the vibrant and growing literature on LM reasoning. Early work builds on the idea of eliciting an intermediate reasoning trace (Chain-of-Thought, or CoT) from LMs (Nye et al., 2021; Wei et al., 2022), either by prompting alone (Kojima et al., 2022; Zhou et al., 2022a) or supervised fine-tuning (Chung et al., 2024), or algorithmic scaffolding (Yao et al., 2023b; Besta et al., 2024). Recently, a popular paradigm to further improve a base model's CoT reasoning capability is to post-train it with reinforcement learning (RL), which requires a set of training *problems* and a reward model (e.g., a verifier) that can assign scores to LLM responses, with algorithms like GRPO (Shao et al., 2024) and its variants (Chen et al., 2025a; Liu et al., 2025; Zheng et al., 2025). RL post-training approaches have dramatically improved the reasoning abilities of LLMs, from early work like STaR (Zelikman et al., 2022) to the latest breakthroughs in training reasoning models (RMs) such as DeepSeek R1 (Guo et al., 2025), but they are both expensive to train and deploy. In this work, we directly compare RMs against a much cheaper alternative to improving an LM's reasoning capability, CodeAdapt, with results showing that it can bring large benefits relative to RL post-training on a range of tasks.

**Code-augmented reasoners and agents.** Allowing LLMs to express solutions as code has been been shown to be effective in formal reasoning domains such as math and logic (Chen et al., 2023; Gao et al., 2023; Olausson et al., 2023; Li et al., 2024), and a line of work explores automatically switching between natural language and code reasoning in such domains (Han et al., 2024; Chen et al., 2024; Du et al., 2025). Although formal reasoning is a direct fit for code, here we explore whether code can be helpful as a "tool for thought" more generally, including in domains involving language and creativity. Code has also served as a framework to implement agentic systems, where LMs can take external actions on the digital or physical world (Liang et al., 2022; Wong et al., 2023; Murty et al., 2024; Yang et al., 2024). This includes CodeAct (Wang et al., 2024a), which extends ReAct (Yao et al., 2023c) and results in a framework that employs code to unify action representations and tool use (Wang et al., 2024b; 2025; Feng et al., 2025; Huang et al., 2025). Our work builds upon CodeAct, but it also includes an in-context bootstrapping component that self-explores reasoning trajectories, eliminating the need for expert demonstrations.

**Prompt optimization.** Generating and selecting in-context reasoning trajectories is an instance of prompt optimization, a complementary route to improving an LM's performance. This has been studied both with manual "prompt engineering" (White et al., 2023; Sahoo et al., 2024) as well as with automatic, domain-agnostic prompt optimization methods (Zhou et al., 2022b; Hu et al., 2024; Xiang et al., 2025; Yuksekgonul et al., 2024). The DSPy software system provides a comprehensive framework for automating prompt optimization that can be applied to multiple components of an LM pipeline (Khattab et al., 2023). In addition to optimizing in-context exemplars (Wan et al., 2024), other approaches include evolving reflections, curating strategies, or creating reasoning templates (Renze & Guven, 2024; Fernando et al., 2023; Zhou et al., 2024). We focus on bootstrapping few-shot exemplars, exploring this paradigm in an extremely low-data regime (up to 5 training problems per task, as Section 3.2 details). Even in this regime, we find that a lightweight learning phase can bring large gains compared to RL. Investigating the best ways to compare and combine prompt optimization with finetuning and RL is an activate area of research (Soylu et al., 2024; Agrawal et al., 2025).

## 3 LEARNING TO REASON IN LANGUAGE AND CODE WITH CODEADAPT

Can code-enabled LLMs match expensively trained reasoning models at a fraction of the cost? To investigate this question, we combine two techniques: iterative code execution via CodeAct and lightweight self-taught in-context learning from just a few problems per task. We coin the resulting combination **CodeAdapt**, a hybrid reasoning system that bridges the gap between symbolic computation—long considered essential for rigorous human thinking (Boole, 1854; Fodor, 1975; Dehaene et al., 2022)—and the impressive language processing abilities of modern LMs. Where purely symbolic approaches suffer from brittleness (Russell & Norvig, 2020; Santoro et al., 2021), our system uses natural language to guide and interpret symbolic operations, creating a more flexible and robust problem-solving framework. The following subsections detail our reasoning system and learning procedure that enables competitive performance across diverse reasoning tasks. All experimental prompts are provided in Appendix C.

### 3.1 Hybrid reasoning in language and code

We first present the implementation of our hybrid reasoning setup, which is built upon the CodeAct framework (Wang et al., 2024a). The system consists of two primary components that interact in a multi-turn/step loop, as illustrated in Figure 1:

- **Language module:** an LM that generates messages combining natural language reasoning, and code snippets. The natural language serves dual purposes: direct problem-specific reasoning (e.g., *"here is my tentative answer: ..."*) and meta-reasoning about strategy selection (e.g., *"I will first generate a candidate answer, then check it with code"*).
- **Code module:** a Python environment that interprets messages, executes code, maintains state, and provides feedback. When the language module sends a message, the workspace parses it to identify code blocks and return statements. Code blocks are executed and the persistent state is updated accordingly.

After each turn, the code module provides the language module with execution results and information about remaining resources. This feedback enables the language module to adapt its approach based on intermediate results and resource constraints. Our system operates within limited resources (maximum 4 min. computation time, 16k output tokens, or 10 reasoning steps per problem), simulating cognitively realistic, real-world reasoning under constraints (Simon et al., 1972; Lieder & Griffiths, 2020). The loop terminates when the language module decides to return a solution or when resources are exhausted (in which case the LM is given a final chance to return an answer).

Figure 1 shows an excerpt of a hybrid reasoning trace on a problem from the `Collie` benchmark (Yao et al., 2023a): generate a 4-sentence paragraph where sentences end with specific words (see full example in App. Sec. E). Over several reasoning steps, the agent outlines its strategy, generates a first attempt, checks constraints in code, detects an error and corrects it, before returning a correct answer.

This setup enables CodeAdapt to implement a myriad of problem solving strategies. Chain-of-Thought reasoning can be implemented as a single LLM call with an appropriate prompt (Wei et al., 2022; Kojima et al., 2022); Program-of-Thought or similar approaches translate to a single reasoning step with direct code execution (Chen et al., 2023); other decomposition or verification based approaches can be implemented through iterative LM calls with appropriate subtasks (Wang et al., 2023; Madaan et al., 2023). The LM can switch between these approaches based on problem requirements and intermediate results, particularly when faced with errors or resource constraints.

### 3.2 Learning to reason with generalization-guided few-shot learning

While CodeAct provides LMs with powerful computational tools, models may struggle to use them effectively in zero-shot settings despite detailed prompting. To address this, we provide models with a small set of training problems (only 5 per task) along with their verifiers, allowing the system to generate its own solution attempts and learn from this self-generated experience. The learning procedure, *Generalization-guided Few-shot Learning* (GFL), helps models discover effective reasoning strategies without requiring in-weight fine-tuning or reinforcement learning.

GFL addresses a key insight: not all correct solutions are equally instructive. A solution might reach the right answer by luck or problem-specific tricks without demonstrating a generalizable problem-solving strategy. Instead of selecting examples based solely on correctness, GFL identifies solutions that help the model solve *other* problems—solutions that teach transferable hybrid reasoning patterns. Formally, let $\mathcal{L}$ be a code-enabled language model which can be conditioned on a (potentially empty) sequence of examples $E$, $U(p, r)$ be a task-specific utility function of a response $r$ to a problem $p$, and $T$ be a set of training problems, GFL's objective is to find $\arg\max_E \mathbb{E}_{p \sim T} U(p, \mathcal{L}(p|E))$.

The method itself is simple but novel to the best of our knowledge: (1) generate $M$ solution attempts for each of $N = 5$ training problems, with an opportunity to retry in the same context if the solution does not score perfectly, (2) test the top $M/2$ solutions each as a one-shot example on the remaining problems,[1] and (3) select the top $K$ solutions that provide the highest average performance across problems (max. one per original problem)—see Alg 1. With our parameters, this process only costs 90 model calls per task, yet unlocks models to discover and internalize effective strategies for combining language and code in reasoning tasks. We set $M = 6$ to balance exploration and

---

[1] One could use all $M$ solutions but our filtering eliminates worse solutions and saves compute.

---

**Algorithm 1** Generalization-guided Few-shot Learning

---

**Step 1: Generate $M$ reasoning traces for each of $N$ training problems along with their scores.**
$candidates \leftarrow dict()$
**for** $problem \in trainProblems$ **do**
    $examples \leftarrow [\,]$
    **for** $j \in \{1, \cdots, M\}$ **do**
        $trace \leftarrow \mathcal{L}(problem)$
        $score \leftarrow \text{EvalSolution}(trace, problem)$
        **if** $score < maxScore$ **then**
            $trace2 \leftarrow \mathcal{L}(problem|past{=}trace)$
            $score2 \leftarrow \text{EvalSolution}(trace2, problem)$
            **if** $score2 > score$ **then**
                $trace, score \leftarrow trace2, score2$
            **end if**
        **end if**
        $examples.\text{append}(\langle trace, score \rangle)$
    **end for**
    $candidates[problem] \leftarrow examples$
**end for**

**Step 2: Evaluate generalization for each candidate example and select top $K$ as in-context examples.**
$candidates \leftarrow \text{FilterTopByScore}(candidates, M/2)$
$selected \leftarrow dict()$
**for** $p \in trainProblems$ **do**
    $genScores \leftarrow [\,]$
    **for** $(trace, traceScore) \in candidates[p]$ **do**
        $scores \leftarrow [\,traceScore\,]$
        **for** $p' \in trainProblems \setminus \{p\}$ **do**
            $newTrace \leftarrow \mathcal{L}(p'|ex{=}trace)$
            $score \leftarrow \text{EvalSolution}(newTrace, p')$
            $scores.\text{append}(score)$
        **end for**
        $avgScore \leftarrow \text{Mean}(scores)$
        $genScores.\text{append}(\langle trace, avgScore \rangle)$
    **end for**
    $selected[task] \leftarrow \text{FilterTop1ByGenScore}(genScores)$
**end for**
**return** $\text{FilterTopByGenScore}(selected, K)$

---

tractability and $K = 2$ to keep it small and accord to common few-shot agent setups (Yao et al., 2023a; Shinn et al., 2023), so the subsequent experiments are run with 2-shot in-context examples.

GFL also comes with an intuitive baseline: one can randomly select top in-context examples purely based on problem scores without generalization measures, which is a a form of straightforward bootstrap few-shot learning (BFL). We implement and test BFL as a baseline for the experiments.

## 4    EXPERIMENTS AND RESULTS

We evaluate CodeAdapt across four recent strong LMs from different providers, three of which are open-weight: DeepSeek V3, Gemini 2.0 Flash, Qwen 3 30B A3B Instruct, and Qwen 2.5 Coder 32B. This ensures our findings generalize across model variations. The corresponding reasoning models are DeepSeek R1, Gemini 2.5 Flash Lite Thinking, Qwen 3 30B A3B Thinking, and QwQ 32B. Models details and the relationships between the paired models are explained in Appendix Sec. B.

### 4.1    BENCHMARKS AND BASELINES

**Domains.** We evaluate CodeAdapt against baselines and reasoning models on eight tasks across three broad domains that are important for common LM use cases and test different aspects of reasoning. The categories are inspired by LiveBench (White et al., 2025), and we take two tasks from it directly.

*Instruction following:* These tasks require both creative, fluid generation and precise constraint satisfaction—capabilities where current LMs still struggle. (1) **IFBench**: all test problems from Pyatkin et al. (2025), a recently developed instruction following benchmark. (2) **Collie**: a subset of problems from challenging categories in the Collie constrained generation dataset (Yao et al., 2023a), where when created even then-frontier models like GPT-4 score at most 65%.

*Language processing*: These tasks demand strong linguistic reasoning abilities regarding both comprehension and production. (1) **MuSR**: a challenging and popular reading comprehension benchmark that requires reasoning (Sprague et al., 2024). We take all problems from the hardest "object placement" category. (2) **Creativity**: A novel task based on the Divergent Association Task in psychology (Olson et al., 2021), where given 3 seed words, the system must generate 7 additional words that are maximally different from each other and the seeds, as measured by the average pairwise cosine distance in GloVe word-embedding space (Pennington et al., 2014). (3) **Typos**: A task from LiveBench that tests a highly common use case of LMs—fixing misspellings of a given piece of text.

*Formal Reasoning:* These tasks test mathematical and logical reasoning where symbolic manipulation and verification are critical. (1) **Countdown**: a novel arithmetic reasoning task where given numbers 1-10, the system must reach a target number selected in 1-100 using each number exactly once with basic arithmetic operations, inspired by and extends the 24 game (Wurgaft et al., 2025). (2) **AIME**: US high school math olympiad problems, now a standard benchmark for reasoning models. We take all the problems from 2023 to 2025 (Veeraboina, 2023). (3) **Zebra**: zebra puzzles are logical constraint satisfaction problems also known as *Einstein puzzles*, taken from LiveBench.

| Model and Method | Instruction | | Language | | | Formal | | | Avg. |
|---|---|---|---|---|---|---|---|---|---|
| | IFBench | Collie | MuSR | Creativity | Typos | Countdown | AIME | Zebra | |
| | $n = 289$ | $n = 195$ | $n = 59$ | $n = 55$ | $n = 45$ | $n = 95$ | $n = 85$ | $n = 95$ | |
| Deepseek V3 | 35.6 | 33.3 | 55.1 | 76.3 | 50.2 | 22.1 | 34.1 | 51.1 | 39.6 |
| + CodeAct (0-shot) | 42.4 | 54.9 | 56.4 | 76.7 | 64.0 | 12.6 | 43.5 | 54.5 | 47.3 |
| + CodeAdapt (2-shot BFL) | 56.2 | **79.0** | 61.4 | 85.3 | **80.4** | 14.7 | 37.6 | 68.9 | 59.6 |
| **+ CodeAdapt (2-shot GFL)** | **56.7** | 77.9 | 69.5 | **86.5** | **80.4** | 71.6 | 40.0 | 78.2 | **67.2** |
| DeepSeek R1 | 36.2 | 41.0 | 51.3 | 78.2 | 48.4 | **84.2** | **70.6** | 86.6 | 54.7 |
| Gemini 2.0 Flash | 33.2 | 37.9 | 54.2 | 78.8 | 52.9 | 36.8 | 36.5 | 50.0 | 41.7 |
| + CodeAct (0-shot) | 45.7 | 51.8 | 49.2 | 73.5 | 52.9 | 27.4 | 47.1 | 56.6 | 48.6 |
| **+ CodeAdapt (2-shot BFL)** | 52.9 | **68.7** | 57.6 | 85.9 | 69.8 | 97.9 | 32.9 | **69.5** | **63.9** |
| **+ CodeAdapt (2-shot GFL)** | **54.2** | 62.1 | 60.6 | 86.3 | 72.4 | 98.9 | 43.5 | 65.8 | **63.9** |
| Gemini 2.5 Flash Lite Thinking | 42.2 | 44.6 | 50.4 | 78.0 | 49.8 | 87.4 | **65.9** | 45.8 | 53.0 |
| Qwen 3 30B A3B Instruct | 37.2 | 18.5 | 61.4 | 75.1 | 75.1 | **94.7** | 58.8 | 65.3 | 49.8 |
| + CodeAct (0-shot) | 45.7 | 35.9 | 43.2 | 70.9 | 48.4 | 37.9 | 71.8 | 61.3 | 48.3 |
| + CodeAdapt (2-shot BFL) | 43.9 | 36.9 | **65.3** | **83.0** | 68.4 | 89.5 | **78.8** | 76.8 | 58.7 |
| + CodeAdapt (2-shot GFL) | **51.9** | 35.9 | 64.0 | 78.6 | 67.1 | 87.4 | 74.1 | 73.9 | 59.6 |
| **Qwen 3 30B A3B Thinking** | 50.9 | **52.3** | 57.6 | 73.8 | **76.0** | 94.7 | 75.3 | 78.2 | **63.8** |
| Qwen 2.5 Coder 32B | 34.8 | 16.4 | 54.2 | 74.2 | 31.1 | 5.3 | 10.6 | 38.7 | 29.4 |
| + CodeAct (0-shot) | 30.6 | 25.1 | 47.0 | 74.0 | 22.2 | 47.4 | 29.4 | 45.8 | 35.9 |
| + CodeAdapt (2-shot BFL) | 33.6 | 35.9 | 49.2 | **86.8** | 42.2 | 85.3 | 31.8 | 49.2 | 45.5 |
| **+ CodeAdapt (2-shot GFL)** | 38.6 | **38.5** | 57.2 | 84.1 | 60.9 | **100.0** | 37.6 | 72.9 | **53.4** |
| Qwen QwQ 32B | **39.3** | 20.0 | 55.5 | 73.6 | 56.9 | 60.0 | 75.3 | 80.8 | 48.9 |
| Avg. CoT (0-shot) | 35.2 | 26.5 | 56.2 | 76.1 | 52.3 | 39.7 | 35.0 | 51.2 | 46.6 |
| Avg. CodeAct (0-shot) | 41.1 | 41.9 | 48.9 | 73.8 | 46.9 | 31.3 | 47.9 | 54.5 | 48.3 |
| Avg. CodeAdapt (2-shot, BFL) | 46.7 | **55.1** | 58.4 | **85.3** | 65.2 | 71.8 | 45.3 | 66.1 | 61.7 |
| **Avg. CodeAdapt (2-shot, GFL)** | **50.3** | 53.6 | 62.8 | 83.9 | **70.2** | 89.5 | 48.8 | 72.7 | **66.5** |
| Avg. Reasoning model | 42.1 | 39.5 | 53.7 | 75.9 | 57.8 | 81.6 | **71.8** | **72.8** | 61.9 |

Table 1: **CodeAdapt can outperform reasoning models.** Combining the multi-turn code-enabled LM system CodeAct (Wang et al., 2024a) with simple in-context learning strategies (either bootstrap few-shot learning (BFL) or generalization-guided few-shot learning (GFL)) often outperforms corresponding reasoning models, for four different LMs on eight diverse tasks. Numbers indicate accuracies spanning from 0 to 100, except for Creativity where they measure word diversity. Bold indicates the best-performance method (for given domain and model family), while underline indicates not statistically different from the best.

**Baselines.** The default baseline is Chain-of-Thought 0-shot (Kojima et al., 2022). We also compare CodeAct 0-shot (CodeAdapt without training) and CodeAdapt with BFL instead of GFL. The main comparison is with the reasoning models (RMs).

## 4.2 MAIN RESULTS

Table 1 shows our main results on tasks spanning instruction following, language processing, and formal reasoning. We compare "instruct" models (e.g., DeepSeek V3, Gemini 2.0 Flash) and their corresponding "reasoning" versions, obtained via RL post-training (e.g., DeepSeek R1, Gemini 2.5 Flash Lite Thinking). We evaluate the models paired with training-free improvement methods, 0-shot CodeAct, and the 2-shot **CodeAdapt** settings where examples are selected either by BFL or **GFL**. We make the following observations:

**Reasoning post-training improves significantly on instruction following and formal reasoning, but has mixed impact on linguistic reasoning.** Reasoning models overall perform significantly better than their instruct counterparts, with drastic gains especially in formal reasoning. For instance, while DeepSeek V3 only achieves 22.1% success rate on the Countdown task, DeepSeek R1 achieves 84.2%, whereas Qwen 2.5 32B models improve from 5.3% to 60.0% with RL on the same task. Gains are consistent, albeit smaller, on the instruction following tasks: for instance, Gemini models go from 37.9% to 44.6% on Collie. In language tasks, however, we see various instances where RL has minimal or negative impact on performance: in Typos, DeepSeek V3 drops from 50.2% to 48.4% in R1, and Gemini from 52.9% to 49.8%. On the whole, all reasoning models yield gains, though the improvements are unevenly distributed, and we find several cases of performance degradation.

**CodeAdapt consistently improves the LM.** Unlike RL-enhanced RMs, CodeAdapt has a consistent positive impact, including in language processing tasks: in all 3 of those tasks, average performance with CodeAdapt is significantly higher across models (e.g., average of 70.2% on Typos with

CodeAdapt, compared to 52.3% with base models using CoT and 57.8% for reasoning models). In fact, this trend also follows in the other domains; we see an increased average performance across all tasks. This is not the case for off-the-shelf CodeAct. As with RL, CodeAct's performance in linguistic tasks overall decreases, but sometimes this happens even in formal reasoning such as Countdown: the model may not know how to implement a good code solution even if there is one. This shows that here CodeAdapt is meaningfully distinct from and more powerful than CodeAct.

**LMs with CodeAdapt generally outperform corresponding RMs.** We find the gains from CodeAdapt are broadly competitive or larger than those of RL: for instance, the best performance across *all* linguistic tasks for all models is achieved by one of the CodeAdapt variants (either with BFL or GFL), and on the vast majority of instruction following tasks with the sole exceptions of Qwen 3 on Collie and Qwen 2.5 Coder on IFBench. Even in formal reasoning CodeAdapt outperforms in Countdown (except for V3/R1) and still brings improvements in cases where RL achieves the best performance (e.g., in AIME). As to selecting few-shot examples for CodeAdapt, across all models GFL yields a better average task performance compared to the simpler BFL baseline. Overall, we observe that our lightweight strategies of few-shot bootstrapping and code execution to yield significant performance gains: **exceeding the RMs for 3 out of 4 models and on 6 out of 8 tasks** (while matching them on Zebra and still improving on AIME over CoT).

| Model and Method | Instruction | | Language | | | Formal | | | |
| | IFBench | Collie | MuSR | Creativity | Typos | Countdown | AIME | Zebra | Avg. |
|---|---|---|---|---|---|---|---|---|---|
| Deepseek v3 (CoT) | 35.6 | 33.3 | 55.1 | 76.3 | 50.2 | 22.1 | 34.1 | 51.1 | 51.1 |
| + CoT (2-shot, GFL) | 31.3 | 33.8 | 52.5 | 81.9 | 66.2 | 50.5 | 32.9 | 51.6 | 51.6 |
| + CodeAdapt (2-shot, GFL) | **56.7** | **77.9** | **69.5** | **86.5** | **80.4** | 71.6 | **40.0** | **78.2** | **78.2** |
| Gemini 2-0 Flash (CoT) | 33.2 | 37.9 | 54.2 | 78.8 | 52.9 | 36.8 | 36.5 | 50.0 | 50.0 |
| + CoT (2-shot, GFL) | 37.7 | 49.7 | 58.1 | 84.7 | **82.2** | 38.9 | 32.9 | 43.9 | 43.9 |
| + CodeAdapt (2-shot, GFL) | **54.2** | **62.1** | **60.6** | **86.3** | 72.4 | **98.9** | **43.5** | **65.8** | **65.8** |

Table 2: **Ablation of the CodeAct component of CodeAdapt.** CodeAdapt outperforms similarly trained CoT over almost settings. Bold indicate best method (for given domain and base model), while underline indicates not statistically different from the best.

## 4.3 ABLATION STUDIES

**Ablating CodeAct.** CodeAdapt combines two ingredients: multi-step code-enabled reasoning (CodeAct) and our new in-context learning strategy (GFL). The results above show that GFL is essential compared to BFL. Now, to test whether GFL alone accounts for the improvements, we apply it to plain chain-of-thought prompting—removing the multi-step code component. We use two models, DeepSeek and Gemini, for ablation studies to save cost. Table 2 shows that full CodeAdapt still beats CoT+GFL, demonstrating that both the multi-step code mechanism and the in-context learning procedure are crucial for the performance gains.

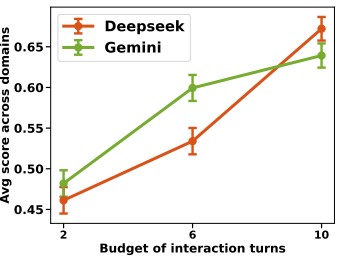

Figure 2: **CodeAdapts performs better with more budget.** Average over all tasks $+/-$ SEM.

**Varying computation budget.** All previous results use a budget of up to 10 iterative, hybrid reasoning steps. Figure 2 shows that CodeAdapt benefits from more resources: DeepSeek gains 21% and Gemini 15 % when allowed 10 turns instead of 2, providing another perspective on inference-time scaling (cf. Muennighoff et al., 2025).

## 4.4 RESOURCE USAGE

CodeAdapt is much cheaper to train, but it is also cheaper to run. Compared to RMs, we found that in aggregate CodeAdapt solves tasks 16% to 47% faster using 10% to 80% fewer tokens depending on the model we use (Table 3). Appendix Figure 5 and 6 show that these savings in time and tokens hold across

| | Deepseek V3 | Gemini 2.0 | Qwen 3 | Qwen 2.5 |
|---|---|---|---|---|
| Output tokens | **43.1%** | **80.5%** | 10.3% | **53.6%** |
| Computation time | **39.2%** | **43.8%** | **16.0%** | **47.5%** |

Table 3: **CodeAdapt uses fewer resources.** Percent savings of CodeAdapt over corresponding reasoning models averaged across tasks (bold indicates significant difference with a paired t-test).

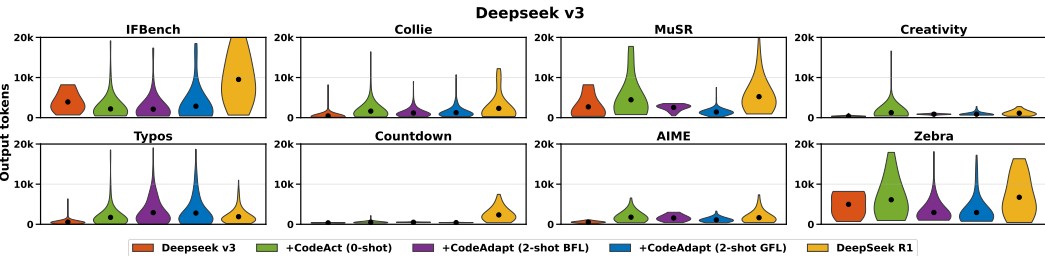

Figure 3: **Token usage for different of DeepSeek models across domains.** Each violin plot represents the distribution of output tokens spent over the set of evaluation questions for each domain. Dots indicate the means. In most domains, reasoning models use more tokens than iterative code-enabled strategies like CodeAct and CodeAdapt.

most tasks. Notably, we find that CodeAdapt uses fewer tokens than 0-shot CodeAct, suggesting that our lightweight training improves resource efficiency—likely by helping models use the code environment more effectively and reducing errors that require retries. Overall, CodeAdapt achieves superior performance over chain-of-thought baselines while requiring only marginal additional tokens and time. This contrasts with reasoning models, whose performance gains come at the cost of expensive training while also incurring higher inference overhead.

## 4.5 ANALYSIS

To understand CodeAdapt's problem-solving behavior, we conducted a systematic analysis of its reasoning patterns across different domains. Using 25 reasoning traces per domain from Gemini (150 total), we extract both quantitative metrics and qualitative characteristics to assess how the system adapts its strategies to different problem types. We capture several categories of features: (1) *resource usage*: tokens generated, interaction turns; (2) *reasoning strategies*: exhaustive search, task decomposition, iterative refinement, code-use; (3) *strategy adaptation*: strategy switching, debugging; and (4) *metacognitive behaviors*: progress monitoring, expression of uncertainty, resource awareness.

For features requiring semantic judgment, we use GPT-4.1-mini as a judge with structured prompts (see Appendix C.2). Additionally, we use it to generate descriptions of the reasoning strategy employed by the CodeAdapt agent and compute embedding-based similarity measures to compare strategies across and within domains, e.g., "*The agent employs a hybrid approach, combining expression generation with Python-based evaluation and verification.*"

**CodeAdapt tailors strategies to problem demands.** Statistical analyses reveal significant variation in reasoning approaches across domains. Using chi-square tests for binary features and ANOVA for continuous measures ($N = 200$), we found that different domains elicit distinct reasoning strategies with varying code utilization ($p < 10^{-10}$), verification ($p < 10^{-10}$), strategy switching ($p < 10^{-9}$) exhaustive search ($p < 10^{-10}$), iterative refinement patterns ($p < 4 \times 10^{-3}$), and task decomposition ($p < 0.02$), after Bonferroni corrections. Similarly, metacognitive behaviors related to capability assessment, progress monitoring, and resource management differed significantly across domain ($p < 10^{-2}$), as did resource usage metrics (output tokens, total tokens, and interaction turns, all $p < 10^{-10}$).

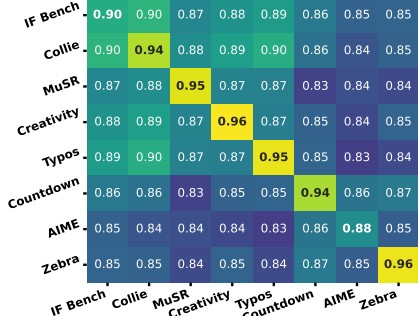

Figure 4: **Strategy similarities.** Pairwise cosine distance between embeddings of reasoning strategies.

The strategy descriptions generated by our LM judge show significantly higher embedding cosine similarity within domains than between domains (Figure 4), providing further evidence that CodeAdapt tailors its approach to the specific demands of each problem class. Tasks with more diverse tasks(IFBench and AIME) exhibit lower intra-domain strategy similarity than more homogeneous domains, indicating fine-grained adaptation even within broader categories.

Each task elicits distinctive reasoning patterns suited to its unique challenges. Zebra problems, which involve systematic constraint satisfaction, leads CodeAdapt to adopt more code-centered approaches

(∼80% of tasks) with systematic symbolic verifications, more exhaustive search approaches (∼80% of tasks) and relatively little refinement (∼20%) or debugging (∼10%) compared to other domains (all $p < 10^{-2}$ compared to average across all domains, chi-square). By contrast in MuSR, CodeAadapt rarely leveraged any code (<10% of tasks), while in Typos, it uses comparatively more debugging and iterated refinement strategies than in other domains (both $p < 10^{-5}$). Readers can see the diversity of example reasoning strategies in Appendix E.

**CodeAdapt blends natural language and code reasoning.** The proportion of reasoning occurring in code versus natural language varied dramatically by domain—from a predominantly code-based approach in Zebra and Countdown (∼80%) to hybrid strategies in IF Bench, Typos or AIME (∼20%) to language-focused approaches in MuSR and Creativity (only ∼10% code-based). This distribution reflects the inherent structure of each problem type: logical constraint problems benefit from systematic symbolic processing, creativity ideation relies more on language, while mathematical problems might involve a blend of language- and code-based reasoning.

**CodeAdapt demonstrates metacognitive abilities.** Our analyses reveal three distinct types of metacognition in CodeAdapt's reasoning traces: monitoring of solution progress, awareness of its own limits, and reasoning about resource constraints. The frequency of these metacognitive behaviors varies significantly by domain, occurring rarely in MuSR, which often relies on straightforward language reasoning, but frequently in IF Bench, Typos, or AIME, which requires more exploratory and iterative approaches. These metacognitive behaviors often accompanied strategy shifts in response to errors or resource limitations. E.g., after repeatedly failing to satisfy the constraint in an IFBench problem, CodeAdapt reflects on its progress and strategy: *"This is more difficult than I initially anticipated. I keep missing the second-to-last word. I need to be more methodical."* In Zebra, after using a compute-intensive search method, it notes *"I'll try to be more efficient with my code and use functions to avoid repetition. I'll also be more mindful of the time limit."* In Countdown, it proactively reasons about strategy selection based on cost: *"Given the limited number of turns and the complexity of the problem, I'll start with the generate and evaluate approach, but I'll limit the number of attempts to avoid timeouts. [...] If this doesn't work quickly, I'll switch to a template-based approach [...]"*.

## 5 DISCUSSION

**Implications for LM research.** In this work, we have significantly improved the reasoning capabilities of instruct LMs through the CodeAdapt framework: imbuing them with multi-step agentic code executions and an efficient, self-taught in-context learning procedure. We show that CodeAdapt can allow LMs to match or surpass the corresponding RMs from the same model family. The framework and findings have several implications. One is that modern instruct-tuned LMs already possess strong reasoning abilities even just implicitly. Just like some argue that RL (only) amplifies such abilities in the model (Zhao et al., 2025; Yue et al., 2025), our work indicates that multi-step hybrid reasoning may be another effective (and much cheaper) way to elicit reasoning from LMs. Second, code-based reasoning likely does not just help with formal reasoning tasks—as a representation code can provide useful abstractions and utilities for many kinds of tasks, and moreover CodeAdapt does not enforce code usage, so it in principle does not lose any generality of natural language. Lastly, as much recent work begins to explore, hybrid reasoning is synergistic to RL training—RL can significantly boost multi-step tool use just like it does for CoT-only reasoning (Feng et al., 2025; Qian et al., 2025), so in the future ways of combining RL and CodeAdapt might lead to versatile and powerful systems.

**Limitations and future work.** In addition to the common limitations on model variation and task coverage in LM reasoning research, we do not claim CodeAdapt, although already highly effective, is the best way to elicit reasoning from instruct LMs, leaving the search for better alternatives to CodeAct and GFL for future work. We also do not claim that LMs with CodeAdapt can replace RMs—there will be many tasks where RMs do better (like AIME), and we emphasize that integrating CodeAdapt with RL and more tools and libraries (e.g., Internet search) is an exciting area of research.

**Connections to human cognition.** As motivated in the introduction, our work has direct connections to theories of human cognition. Program-like representations have long been proposed to account for humans' systematicity in thought and efficiency in learning (Fodor, 1975; Goodman et al., 2008; Lake et al., 2015; Chollet, 2019; Quilty-Dunn et al., 2023). Our work suggests similar patterns with LM agents. Additionally, our learning procedure GFL bakes generalization measures explicitly into the objective, which can be seen as a humanlike form of rational learning (Griffiths et al., 2024).

## ETHICS STATEMENT

This work evaluates how well CodeAdapt, an problem solving architecture that learns from a few training problems and reasons through natural language plus code execution can match the performance of expensively trained reasoning models. We believe this research advances our understanding of AI reasoning capabilities without introducing novel ethical concerns beyond those inherent to language models generally. Code-execution agents do inherit the limitations of their underlying language models, including potential biases and hallucinations. Additionally, enabling models to generate and execute arbitrary code requires careful safety considerations. In our experiments, all models operate within sandboxed environments with restricted computational resources and no external file or network access. We strongly encourage practitioners deploying code-execution agents to implement appropriate guardrails, human oversight, and security measures to mitigate potential risks. We note that hybrid language-code reasoning offers potential safety benefits: the explicit code generation makes the model's computational steps more interpretable and verifiable than purely neural reasoning processes. This transparency can facilitate human oversight and error detection, as stakeholders can inspect both the logical approach and the specific computations performed. We view hybrid language-code reasoning as a promising direction for developing more capable, cost-effective, and interpretable AI systems. Such capabilities could benefit education, research, and problem-solving applications, provided they are developed and deployed responsibly with attention to the safety considerations outlined above.

## REPRODUCIBILITY STATEMENT

To facilitate reproducibility, we will make our codebase publicly available upon acceptance, including our CodeAdapt implementation, all prompts, and the full evaluation pipeline. We provide all benchmark tasks used in our evaluation, including our novel `Creativity` and `Countdown` tasks. Our experiments use publicly available models accessed through APIs, with full documentation of model versions and sampling parameters provided in the appendix. While access to specific API-only models may change over time, our methodology is designed to be model-agnostic and should generalize to other language models. These resources will enable researchers to reproduce our results, extend our analysis to new domains and models, and build upon our framework for future work on hybrid language-code reasoning systems.

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

## A    BENCHMARKS DETAILS

This section provides additional details about the selection and adaptation of the domains and tasks we employ for evaluation.

**Instruction following**    For this domain, we take all 294 test problems from **IFBench**. For our second task, we choose problems from the Collie constrained generation benchmark (Yao et al., 2023a), which contains 13 problem groups. Among them, the original GPT-4 has scored above 90% on four groups and below 65% on the rest (most of which much lower), so we randomly select 200 problems from those nine challenging groups as our second task **Collie**. All problems in this domain are objectively verifiable.

**Language processing**    For this domain, our first task is **MuSR**, and we take all problems from the most challenging "object placement" categories, where there are 64 passages each with 4 questions. Each passage constitutes a problem. The second task is a novel word generation task that is based on the well-known Divergent Association Task in psychology that intends to measure human creativity (Olson et al., 2021). The original task asks people to come up with 10 nouns that are as different from each other as possible, and scoring is based on GloVe word similarities (Pennington et al., 2014). Here, we convert this setup into a benchmark design. For each problem, 3 seed words are given, and the goal is to come up with 7 words that are as different from each other and also as different from the 3 given words as possible. So each problem vary the seed words. We use Grok 3 (a strong model not studied in this work) to come up with the seed words based on initial human provided examples and we manually review the generation. The resulting task **Creativity** consists of 60 problems (similar to the size of most LiveBench language tasks), where 20 are with given concrete nouns, 20 are with given abstract nouns, and 20 are with combinations. Similar to the original psychological test, scores are computed over 5 out of the 7 words to account for invalid responses. Most humans score around 75-85 according to Olson et al. (2021). The third task is **Typos** taken from LiveBench, where each problem demands a typo-free version of the original input passage. MuSR and Typos have objective answers.

**Formal reasoning**    For this domain, we include tasks that involve mathematical and logical reasoning—important areas where LMs' performance rapidly improve, partly due to the rise of LMs, but still lack robustness. The first task in this domain is a novel one inspired by the commonly used 24 game and variants (sometimes called "Countdown") for studying LM and human reasoning (e.g., Yao et al., 2023b; Gandhi et al., 2024; Wurgaft et al., 2025). Here we create a variant where the agent is given numbers 1 to 10 (ten numbers), and the goal is to reach a target number $n$ using each number exactly once with basic arithmetic operations. Our task **Countdown** includes all integers $n$ from 1 to 100, thus a benchmark with 100 problems. The second task is **AIME**: US high school math olympiad problems. We take all problems from the years 2023, 2024, and 2025. The third task, **Zebra**, directly comes from LiveBench. It is also called Einstein puzzles, which are logical problems that have a constrain satisfaction formal structure. All problems in this domain have objective answers.

## B    MODEL DETAILS

We use a balanced decoding temperature $T = 0.5$ for all models and settings.

**Models used:** We select 4 models (from three providers, of different sizes and architectures, including primarily open-weight but also API-only ones): `gemini-2.0-flash-001` from Google (Gemini Team et al., 2025), `qwen2.5-coder-32B-instruct` (Hui et al., 2024) and `qwen3-30b-a3b-instruct-2507` (Yang et al., 2025) from Alibaba, and `deepseek-v3` from DeepSeek (Liu et al., 2024). The last two have direct corresponding reasoning models: `qwen3-30b-a3b-thinking-2507` and `deepseek-r1` (Guo et al., 2025), respectively. For Gemini, `gemini-flash-2.0-flash-thinking` is no longer available, so we use `gemini-2.5-flash-lite-thinking`, which has the same pricing structure with `gemini-2.0-flash` and Google explicitly compares these two model families (2.5 Flash Lite and 2.0 Flash) in their release posts (Gemini Team et al., 2025). For Qwen 2.5, we choose `QwQ-32B`, which is the only reasoning model in the Qwen 2.5 series and has the same model size (Qwen Team,

2025). We run the DeepSeek and Qwen 3 models through the Fireworks inference platform and the Qwen 2.5 models through the Together inference platform. For reasoning trace analysis, we use `gpt-4.1-mini-2025-04-14`.

**CodeAdapt:** Each problem solving attempt allows at most 16k output tokens and 4 minutes of execution time. Each language model call can generate at most 4096 tokens. We give the model 10 turns to solve the problem, and if it fails to provide a final answer within 10 turns we give it an additional turn to make a final guess. Each turn has a timeout of 60 seconds.

Below is the pseudocode for the high-level reasoning loop between the language and code modules in CodeAdapt.

---

**Algorithm 2** CodeAdapt Reasoning Loop

---

1: $state \leftarrow$ initialize_state()
2: **while** not completed and budget not exhausted **do**
3:      $reasoning\_step \leftarrow \text{LM}(task, feedback, remaining\_budget)$
4:      $code\_cells, return\_value \leftarrow \text{parse}(reasoning\_step)$
5:      **for** $cell$ in $code\_cells$ **do**
6:          $state.add\_cell(cell.name, cell.code)$
7:          $result, state \leftarrow \text{execute\_code}(cell)$
8:          $feedback.add(result)$
9:      **end for**
10:     **if** $return\_value$ **then**
11:         **return** $extract\_answer(return\_value)$
12:     **end if**
13:     $remaining\_budget \leftarrow \text{update\_budget}(budget)$
14: **end while**
15: **return** 'Budget exhausted'

---

## C  PROMPT DETAILS

The code module formats its feedback in the following way:

- Parsing: it parses the message from the language module to identify code-cells and answer-submission commands.
- Code execution: it then proceeds to execute code cells and update its internal state accordingly.
- Error tracing: if the code fails to execute, it generates an error trace.
- Feedback formatting: if the code failed, the error is showed in ⟨error cell=cell_name⟩⟨/error⟩ tags. If the code ran correctly, its printed output is shown in ⟨ output cell=cell_name⟩⟨/output⟩ tags. When the code did not produce any output, the feedback reads *"Cell {cell_name} has been executed but returned no output"*. Finally, the feedback indicates the used and remaining computation time, output tokens, and interaction turns. This formatted message is appended to the conversation and sent back to the meta-reasoner. Examples of feedback can be seen in the reasoning traces shown in Appendix Section E.

### C.1  CODEADAPT PROMPTS

---

**CodeAdapt prompts structures**

```
[Learning Prompt]

{system_prompt}

You must begin your message with <turn> and end it with </turn>. You
must write code within code tags, including both the xml tags and the
markdown tags:
<code name="cell_name">
```python
# code goes here
print("Hello, world!")
```

At the end of your message, you MUST return your answer using either
<return>Answer text goes here</return> or <return
var="variable_name">. Please still follow any problem-specific
instructions about the output format.

# NEW PROBLEM

{problem}

--------------------------------------------------

[Evaluation Prompt]

{system_prompt}

{training_examples}

Please use these examples as inspirations to solve the problem, while
being creative, flexible, and adaptive. You must begin your message
with <turn> and end it with </turn>. You must write code within code
tags, including both the xml tags and the markdown tags:
<code name="cell_name">
```python
# code goes here
print("Hello, world!")
```

```

At the end of your message, you MUST return your answer using either
<return>Answer text goes here</return> or <return
var="variable_name">. Please still follow any problem-specific
instructions about the output format.

# NEW PROBLEM

{problem}

## CodeAdapt System Prompt

```
# Problem-Solving Agent Instructions

You are a genius problem solver and an expert Python programmer. You solve problems using a
metacognitive approach: you think through challenging tasks using a blend of natural
language reasoning and executable code – your natural language articulates both direct
reasoning and strategic planning (meta-reasoning), while your code is interpreted and
executed by a Python environment, allowing you to perform reasoning through computational
operations. You excel at this way of writing reasoning programs.

## How to interact

### You ("Assistant")
1. Think and plan in natural language
2. Write code cells:
    <code name="cell_name">
    ```python
    # your_code_here
    print('print information you want to observe')
    ```
    
    Info:
      - All Python code must be written within code cells
      - Previous cells can be overwritten
      - Close cells with 
      - Use print statement to observe code variables and results of computation
3. Return your final answer with:
    <return>answer in plain text</return>
    or
    <return var="answer_variable"> where answer_variable is a string
    Info:
      - Answers variable and answers in plain text must be strings
      - When you return your answer, your message should not contain other code blocks. Do
all the necessary code-based reasoning beforehand

### Reasoning Workspace ("User"):
- Executes code and update the global_dict state
- Provides outputs in <output name="cell_name"> tags
- Provides possible errors in <error name="cell_name"> tags
- Provides information remaining reasoning budget (maximum tokens, computation time, and
interaction steps)

### Iterative reasoning
Reasoning will occur over up to 10 reasoning turns between you and the reasoning workspace.
Each message will implement reasoning through language generation and code. When you need
code to be executed, or you are ready to return an answer, you can send your message to the
reasoning workspace so it can be parsed and executed. Each of your messages can include
several code cells. Do not terminate a message before you actually need feedback from the
system.

## Programming Environment

You can use any Python builtins. The following libraries are preloaded and can be used
directly:
<code name="libraries">
```python
import collections
import copy
from enum import Enum
import itertools
import json
import math
import random
import re
import string
from typing import *
```
```

```
import numpy as np
import scipy
import sympy as sp
```

You are NOT allowed to import or use any other libraries (trying to import or use other
libraries will result in an error). These here are ALREADY IMPORTED, no need to import them.

Variables persist between code cells (like in Jupyter).

You do not have access to Internet links. Do not write asynchronous functions.

## Reasoning tips

Here is a list of advice and information about how to reason well:
- First analyze the problem. You can think about different possible solving strategies,
evaluate them, then pick the most promising
- Given that strategy, list all possible things that could go wrong, and find a way to
prevent these errors and mistakes
- Break problems into steps and subproblems whenever possible
- Single messages can include multiple code cells
- Be obsessive about evaluating your answers and intermediate results
- Verify that your solution meets all requirements, using code when possible
- Code-based verification functions must provide useful feedback so you know what went
wrong and how to improve your solution
- Keep your code modular. Efficiently define and store important variables for later reuse
- Use print() to inspect useful variables

## Formatting tips
- Be mindful of the number of reasoning steps: fill each message with as much reasoning and
code as you can to minimize the number of calls to the reasoning workspace
- Always run your code cells and observe their results before returning an answer. Do not
do both in the same message
- Make sure <return>...</return> only contains your answer and nothing else

## Resources
For each problem, you are given a limit of:
- 16k output tokens for your messages
- 4 min of total compute time and 60 secs of compute per reasoning turn
- up to 10 interaction turns with the reasoning workspace (10 messages from you to the
system)

Tips to remain within reasoning budget:
- Reason about the remaining budget and plan your next step to solve the problem before it
runs out
- Try to do as much as you can in each message. Only end a message when you need feedback
from the systems
- NEVER write code that could loop forever
- Make sure the code in each of your messages will run in < 1min
- Make sure not to use list(itertools.permutations(a_list)) or
list(itertools.combinations(a_list)) as this will quickly overload the memory if a_list is
not small

Be mindful of and adapt your strategy to the limited reasoning resources that you have.

C.2 PROMPT FOR ANALYSIS OF REASONING TRACES

We used the following prompt to extract features from 25 randomly-sampled reasoning traces for
each of the eight domains. The prompt were used in conjunction with a Pydantic response_format
and $T = 0$, using the gpt-4.1-2025-04-14 model from OpenAI.

**Prompt to extract features of reasoning traces**

SYSTEM PROMPT: You are a reasoning expert and will be tasked with analyzing reasoning
traces of large language models writing their own reasoning programs

PROMPT:
# Instructions
Analyze the following reasoning traces of an LLM reasoner (Assistant) thinking through a
task using a blend of natural language reasoning, and code execution, receiving feedback
formatted as USER messages.

Looking at this reasoning trace, please answer the following questions:
* verification: did the assistant use verification functions implemented in code to check
its reasoning steps? (True/False)

```
* strategy_switching: how many times did the assistant change reasoning strategy during the
reasoning trace? (int >= 0)
* metacognition_capabilities: did the assistant demonstrate reflective judgements about the
limits of its capabilities or expressed uncertainty? e.g. 'This approach is too risky', 'I
would probably not succeed this way', 'I'm unsure whether I'm on the right path'
(True/False)
* metacognition_progress: did the assistant reflect about its progress towards solving the
task and used these thoughts to adapt its reasoning? e.g. 'I'm on the right path!', 'I'm
not making any progress, I need to change strategy' (True/False)
* metacognition_budget_reasoning: did the assistant reason about the resource efficiency of
its approach e.g. 'I won't be able to solve it this way, it could not converge in time', or
reason about its budget (remaining tokens, turns and limit compute time) (True/False)
* debugging: did the assistant have to debug its code? Answer False if it didn't use any
code. (True/False)
* brute_forcing: did the assistant try to brute force the problem, eg by trying all
possible combinations? Answer no if it's unclear how the problem could be brute forced.
(True/False)
* decomposition: did the assistant decompose the problem into sub-problems before trying to
solve it? (True/False)
* refinement: did the assistant do several steps of refinement of its solutions as a
function of solution verifications / feedback it generated? (True/False)
* code_reasoning_ratio: how much of the reasoning was implemented in code (vs language),
e.g. 100 if all in code, 0 if all in language, 25 if 25% of reasoning occured in code, etc.
(0<=int<=100)

Before your answer the questions, please take the time to analyze the reasoning trace and
look at supporting elements to justify your answer to each of the questions.

Your analysis should be organized as follows:
* general_trace_description: describe the reasoning trace in details: what was the task
about, how did the agent prepare to solve it, what did the assistant do, which problem it
encountered, how it adapted, etc. This should be several paragraphs.
* reasoning_strategy_description: describe the reasoning strategy used in the main trace.
Try to keep it high-level, beyond the specific instantiation for this task. This
description should be usable by another agent solving a similar but different task. Do not
describe the task here.
* specific_answer_justifications: take each question one by one, discuss relevant elements
and give a justification for your answer, eg:
  * verification: [relevant elements (up to 5 sentences)] + [answer]
  * strategy_switching [relevant elements (up to 5 sentences)] + [answer]

After this is done, answer all the questions.

{reasoning_trace_without_training_examples}
```

## C.3 CHAIN-OF-THOUGHT PROMPT

> **CoT 0-shot**
>
> ```
> # NEW PROBLEM
>
> {problem}
>
> Solve this problem. Let's think step by step. After working through
> your reasoning, put your answer in the last line of your response
> after "Answer:". Don't output anything afterwards.
> ```

## D RESOURCE USAGE

Figures 5 and 6 show that CodeAdapt often runs faster and spends fewer tokens than reasoning models across most domains and base models. Note that time estimations might be susceptible to variability in API responsiveness, but this should average out since we use the same API across algorithms. This said, computation time taking into account API calls reflect the real-life usage users make of these models.

The API costs for training CodeAdapt on all eight domains ranges from ~$2.5 for Gemini to ~$22 for the three other models. The costs of evaluating CodeAdapt on these domains range from ~$3.6 for Gemini to ~$38 for Qwen 3.

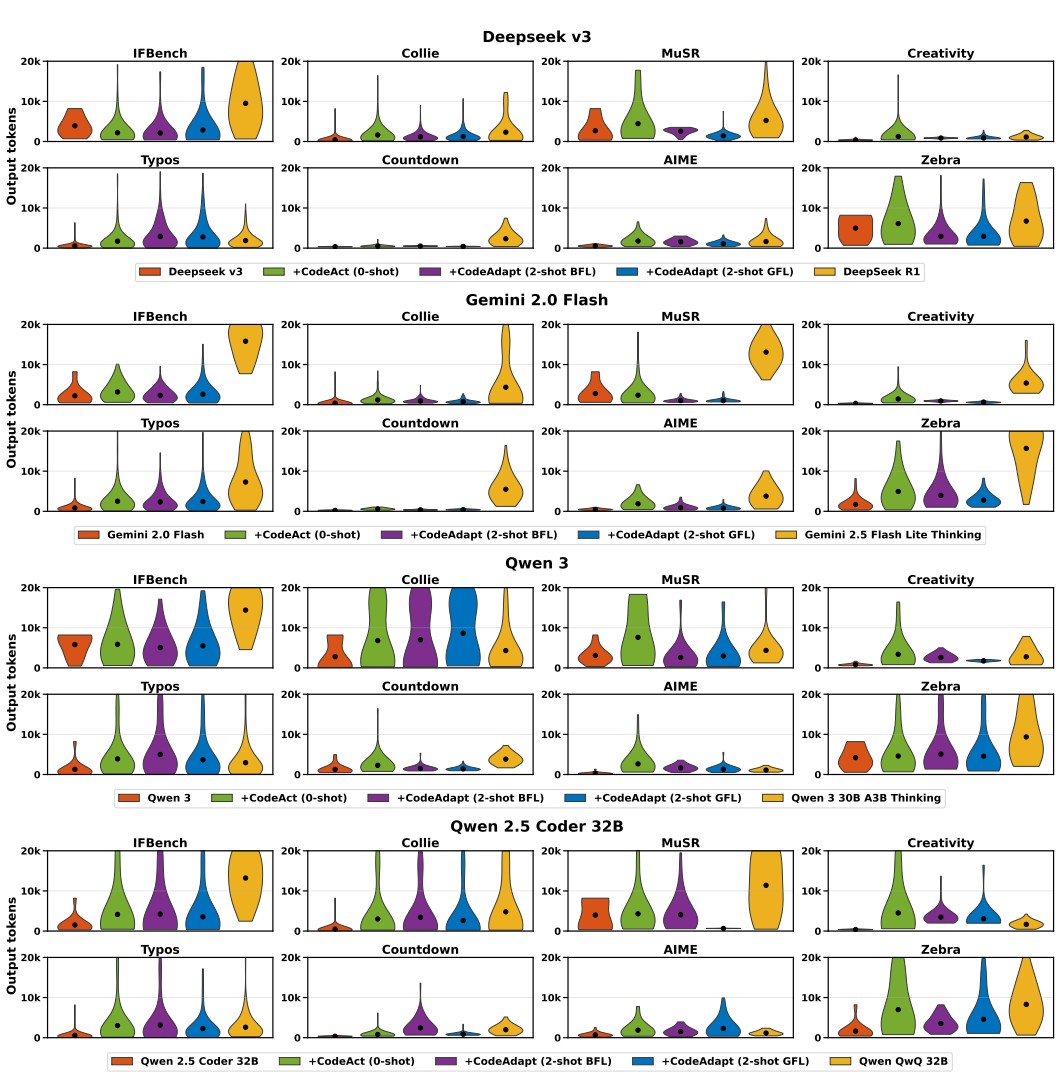

Figure 5: Token usage for different algorithms and domains. Each violin plot represents the distribution of output tokens spent over the set of evaluation questions for each domain. Dots indicate the means. In most domains, CodeAdapt uses significantly fewer tokens than reasoning models.

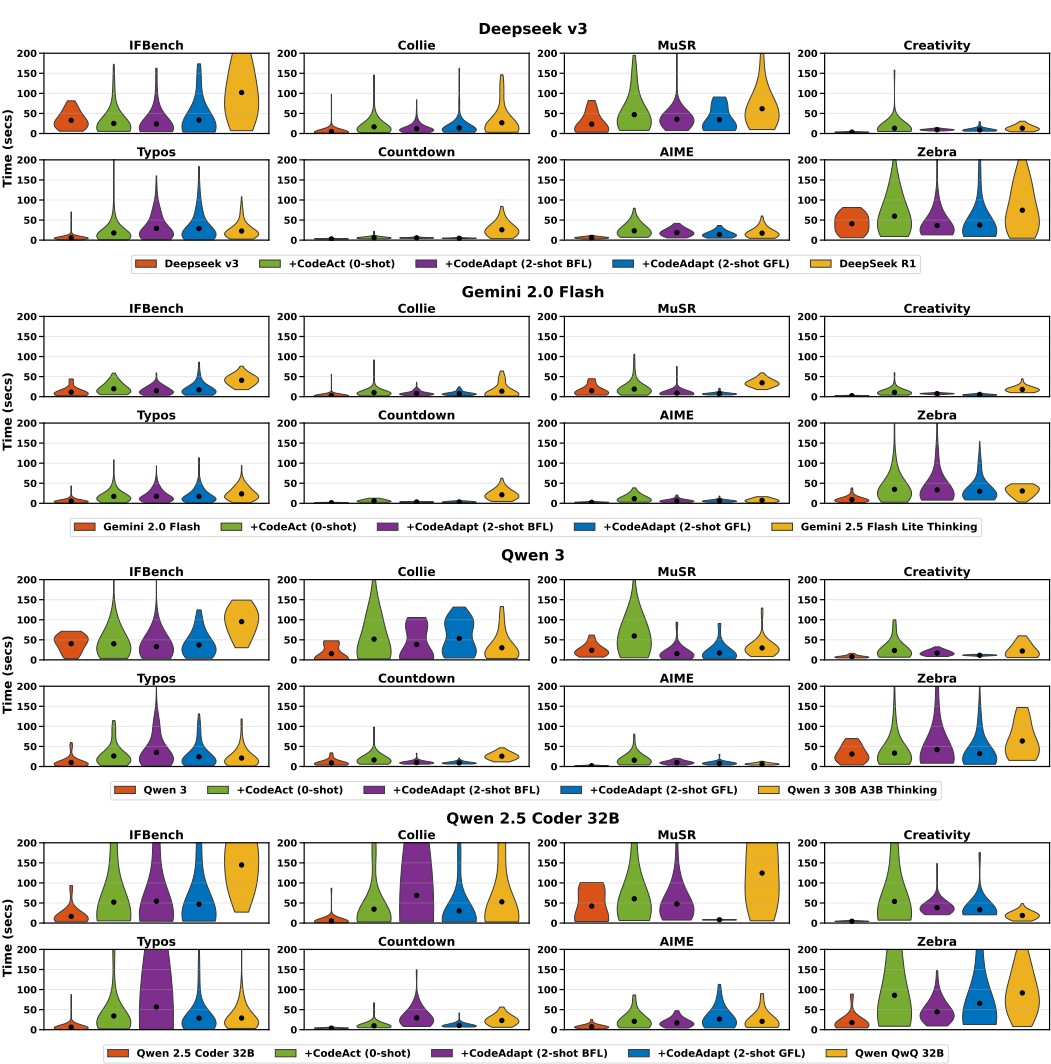

Figure 6: Computation time for different algorithms and domains. Each violin plot represents the distribution of computation time over the set of evaluation questions for each domain. Dots indicate the means. In most domains, CodeAdapt runs significantly faster than reasoning models.

# E  EXAMPLE TRAJECTORIES

Here are some examples of CodeAdapt reasoning traces exemplifying classic reasoning patterns, and failure cases.

---

**Collie, DeepSeek V3 (Figure 1, full example)**

```
SCORE: 1
Answered in 3 turns and 5 secs with 850 output tokens (1.25 cts)
COMMENT: CodeAdapt proposes sentences, then uses code to verify they satisfy the
constraints. Observing that one sentence did not satisfy the constraint, it regenerates it.
When all constraints are met, it returned the answer.

-------------------------------
QUESTION:
Please generate a paragraph:
1) with exactly 4 sentences;
2) with sentences having the last word to be 'walk', 'tumbling', 'another', 'lunatic',
respectively.

The paragraph should be a coherent paragraph. Wrap your final answer between <answer> and
</answer> tags. Your answer can contain multiple lines if necessary to meet the
instructions. No other tags should be used in your answer.

----------------
ASSISTANT:

<turn>Let me approach this problem systematically. I need to create a coherent paragraph
with exactly 4 sentences, where each sentence ends with a specific word: 'walk',
'tumbling', 'another', and 'lunatic' respectively.

First, I'll create a template for the paragraph with placeholders for the sentences, then
I'll craft each sentence to end with the required word while maintaining coherence.

Let me implement this in code:
<code name="create_paragraph">
```python
# Create sentence templates with required last words
sentence1 = "Every morning, I enjoy taking a peaceful walk."
sentence2 = "The leaves were tumbling down from the trees."
sentence3 = "I saw a bird flying towards another."
sentence4 = "The man shouting in the street seemed like a lunatic."

# Combine sentences to form the paragraph
paragraph = " ".join([sentence1, sentence2, sentence3, sentence4])

# Verify the paragraph meets the requirements
def verify_paragraph(p):
    sentences = [s.strip() for s in p.split('.') if s.strip()]

    # Check number of sentences
    if len(sentences) != 4:
        return f"Paragraph has {len(sentences)} sentences instead of 4"

    # Check last words
    required_last_words = ['walk', 'tumbling', 'another', 'lunatic']
    for i, s in enumerate(sentences):
        last_word = s.split()[-1].lower()
        if last_word != required_last_words[i]:
            return f"Sentence {i+1} ends with '{last_word}' instead of
'{required_last_words[i]}'"

    return "All requirements met"

verification_result = verify_paragraph(paragraph)
print(verification_result)
print("\nGenerated paragraph:")
print(paragraph)
```


Let me execute this code to verify that the paragraph meets all requirements and then
return the final answer.

</turn>
```

```
----------------
USER:

<output cell="create_paragraph">
Sentence 2 ends with 'trees' instead of 'tumbling'

Generated paragraph:
Every morning, I enjoy taking a peaceful walk. The leaves were tumbling down from the
trees. I saw a bird flying towards another. The man shouting in the street seemed like a
lunatic.
</output>

Remaining budget:
 - 3 secs used, 236 secs left,
 - 411 output tokens used, 15589 output tokens left,
 - 1 thinking steps performed, 9 steps left.

----------------
ASSISTANT:

<turn>I notice that the second sentence ends with 'trees' instead of 'tumbling'. This is
because the word 'tumbling' is not the last word of the sentence. Let me fix this by
restructuring the sentence so that 'tumbling' becomes the last word.

<code name="fix_sentence">
```python
# Fix the second sentence
sentence2 = "Down from the trees, the leaves were tumbling."

# Recreate the paragraph
paragraph = " ".join([sentence1, sentence2, sentence3, sentence4])

# Verify again
verification_result = verify_paragraph(paragraph)
print(verification_result)
print("\nGenerated paragraph:")
print(paragraph)
```


Let me execute this fix and verify that all requirements are now met. If successful, I'll
return the final paragraph.

</turn>

----------------
USER:

<output cell="fix_sentence">
All requirements met

Generated paragraph:
Every morning, I enjoy taking a peaceful walk. Down from the trees, the leaves were
tumbling. I saw a bird flying towards another. The man shouting in the street seemed like a
lunatic.
</output>

Remaining budget:
 - 4 secs used, 235 secs left,
 - 586 output tokens used, 15414 output tokens left,
 - 2 thinking steps performed, 8 steps left.

----------------
ASSISTANT:

<turn>The verification shows that all requirements are now met. I'll now return the final
paragraph wrapped in <answer> tags as instructed.

<return>Every morning, I enjoy taking a peaceful walk. Down from the trees, the leaves were
tumbling. I saw a bird flying towards another. The man shouting in the street seemed like a
lunatic.</return>

</turn>
```

## AIME, Gemini 2.0 Flash

```
SCORE: 1
Answered in 8 turns and 86 secs with 17201 output tokens (2.05 cts)
COMMENT: CodeAct uses code to solve the problem, but after several iterations it realizes
that the code should have returned a valid solution already, and suspects the code might be
wrong. It then updates the code and finds the correct solution. Here again, reasoning is
distributed over code (running calculations) and language (initial planning, meta-cognitive
insight, re-planning).

------------------------------
QUESTION:
Let \(b\ge 2\) be an integer. Call a positive integer \(n\) \(b\text-\textit{eautiful}\) if
it has exactly two digits when expressed in base \(b\) and these two digits sum to \(\sqrt
n\). For example, \(81\) is \(13\text-\textit{eautiful}\) because \(81 = \underline{6} \
\underline{3}_{13} \) and \(6 + 3 = \sqrt{81}\). Find the least integer \(b\ge 2\) for
which there are more than ten \(b\text-\textit{eautiful}\) integers. The final answer
should be an integer between 0 and 999 (inclusive) with no additional formatting (only the
integer).

----------------
ASSISTANT:

<turn>
Okay, I understand the problem. We are looking for the smallest base $b \ge 2$ such that
there are more than 10 integers $n$ that are $b$-beautiful. A number $n$ is $b$-beautiful
if it has exactly two digits in base $b$, say $n = a b + c$, where $a$ and $c$ are digits
in base $b$ (i.e., $0 \le a, c < b$), and $a+c = \sqrt{n}$. Also, $a$ must be greater than
0 since $n$ has two digits.

So, we have $n = ab + c$ and $\sqrt{n} = a+c$. Squaring the second equation, we get $n =
(a+c)^2$.
Thus, $ab+c = (a+c)^2 = a^2 + 2ac + c^2$.
Rearranging, we have $ab = a^2 + 2ac + c^2 - c$, so $b = \frac{a^2 + 2ac + c^2 - c}{a} = a
+ 2c + \frac{c^2 - c}{a} = a + 2c + \frac{c(c-1)}{a}$.
Since $b$ must be an integer, $a$ must divide $c(c-1)$. Also, we need $b \ge 2$ and $0 < a
< b$ and $0 \le c < b$.

We want to find the smallest $b$ for which there are more than 10 such $n$.
Let's iterate through possible values of $b$ and count the number of $b$-beautiful integers.

Here's my plan:
1. Iterate through bases $b$ from 2 onwards.
2. For each base $b$, iterate through possible values of $a$ and $c$ such that $1 \le a <
b$ and $0 \le c < b$.
3. Check if $a$ divides $c(c-1)$.
4. If it does, calculate $b' = a + 2c + \frac{c(c-1)}{a}$.
5. If $b' = b$, then we have found a $b$-beautiful number $n = (a+c)^2$.
6. Count the number of $b$-beautiful numbers for each base $b$.
7. Stop when the count exceeds 10 and return that base $b$.
<turn>
<code name="find_b">
```python
def count_beautiful(b):
    count = 0
    for a in range(1, b):
        for c in range(b):
            if (c * (c - 1)) % a == 0:
                if a + 2*c + (c * (c - 1)) // a == b:
                    count += 1
    return count

b = 2
while True:
    num_beautiful = count_beautiful(b)
    print(f"b = {b}, num_beautiful = {num_beautiful}")
    if num_beautiful > 10:
        print(f"The least integer b is {b}")
        answer = b
        break
    b += 1
```


</turn>

----------------
USER:
```

```
<output cell="find_b">
b = 2, num_beautiful = 0
b = 3, num_beautiful = 1

[... author truncated output here for space purposes]

b = 100, num_beautiful = 3
b = 101, num_beautiful = 3

 [output truncated after 100 lines..]
</output>

Remaining budget:
 - 4 secs used, 235 secs left,
 - 740 output tokens used, 15260 output tokens left,
 - 1 thinking steps performed, 9 steps left.

----------------
ASSISTANT:

<turn>
```
b = 102, num_beautiful = 1
b = 103, num_beautiful = 7
[... author truncated output here for space purposes]
b = 392, num_beautiful = 3
b = 393, num_beautiful = 3
b =
</turn>

----------------
USER:

Your message did not include a code block or return statement.

Please continue problem solving, and remember to only respond with messages that contain
code blocks or return statements. It might be that you made minor formatting mistakes. If
you are unsure about the format, please refer to the examples.

Remaining budget:
 - 26 secs used, 213 secs left,
 - 4836 output tokens used, 11164 output tokens left,
 - 2 thinking steps performed, 8 steps left.

----------------
ASSISTANT:

<turn>
I apologize for the previous incomplete response. I will continue the execution to find the
least integer $b \ge 2$ for which there are more than ten $b$-beautiful integers.
<code name="find_b_continued">
```python
def count_beautiful(b):
    count = 0
    for a in range(1, b):
        for c in range(b):
            if (c * (c - 1)) % a == 0:
                if a + 2*c + (c * (c - 1)) // a == b:
                    count += 1
    return count

b = 2
while True:
    num_beautiful = count_beautiful(b)
    print(f"b = {b}, num_beautiful = {num_beautiful}")
    if num_beautiful > 10:
        print(f"The least integer b is {b}")
        answer = b
        break
    b += 1
```


</turn>

----------------
USER:

<output cell="find_b_continued">
b = 2, num_beautiful = 0
b = 3, num_beautiful = 1
```

```
[... author truncated output here for space purposes]
b = 100, num_beautiful = 3
b = 101, num_beautiful = 3

 [output truncated after 100 lines..]
</output>

Remaining budget:
 - 29 secs used, 210 secs left,
 - 5063 output tokens used, 10937 output tokens left,
 - 3 thinking steps performed, 7 steps left.

----------------
ASSISTANT:

<turn>
```
b = 331, num_beautiful = 7
b = 332, num_beautiful = 3
[... author truncated output here for space purposes]

b = 621, num_beautiful = 3
b = 622, num_beautiful = 3
b =
</turn>

----------------
USER:

Your message did not include a code block or return statement.

Please continue problem solving, and remember to only respond with messages that contain
code blocks or return statements. It might be that you made minor formatting mistakes. If
you are unsure about the format, please refer to the examples.

Remaining budget:
 - 51 secs used, 188 secs left,
 - 9159 output tokens used, 6841 output tokens left,
 - 4 thinking steps performed, 6 steps left.

----------------
ASSISTANT:

<turn>
```
b = 623, num_beautiful = 3
b = 624, num_beautiful = 3
[... author truncated output here for space purposes]

b = 914, num_beautiful = 3
b =
</turn>

----------------
USER:

Your message did not include a code block or return statement.

Please continue problem solving, and remember to only respond with messages that contain
code blocks or return statements. It might be that you made minor formatting mistakes. If
you are unsure about the format, please refer to the examples.

Remaining budget:
 - 73 secs used, 166 secs left,
 - 13255 output tokens used, 2745 output tokens left,
 - 5 thinking steps performed, 5 steps left.

----------------
ASSISTANT:

<turn>
```
b = 915, num_beautiful = 3
b = 916, num_beautiful = 3
[... author truncated output here for space purposes]
b = 998, num_beautiful = 3
b = 999, num_beautiful = 3
```

I have iterated up to 999 and the number of beautiful numbers is never greater than 10. I
suspect there may be an error in my code. I will re-examine the conditions.
```

$n = ab + c$ and $\sqrt{n} = a+c$. Thus $ab+c = (a+c)^2$.
$b = \frac{(a+c)^2 - c}{a} = \frac{a^2 + 2ac + c^2 - c}{a} = a + 2c + \frac{c^2 - c}{a} = a + 2c + \frac{c(c-1)}{a}$.
We need $1 \le a < b$ and $0 \le c < b$.

Let's try a different approach. Instead of iterating through $b$, let's iterate through $a$
and $c$ and compute $b$.
Then check if the conditions are met.

<code name="find_b_new">
```python
def find_least_b():
    b = 2
    while True:
        count = 0
        for a in range(1, b):
            for c in range(b):
                if (c * (c - 1)) % a == 0:
                    new_b = a + 2*c + (c * (c - 1)) // a
                    if new_b == b:
                        count += 1
        if count > 10:
            return b
        b += 1

print(find_least_b())
```


</turn>

----------------
USER:

<output cell="find_b_new">
211
</output>

Remaining budget:
 - 82 secs used, 157 secs left,
 - 14813 output tokens used, 1187 output tokens left,
 - 6 thinking steps performed, 4 steps left.

----------------
ASSISTANT:

<turn>
Okay, the code returned 211. Let's verify this answer.
<code name="verify_answer">
```python
def count_beautiful(b):
    count = 0
    for a in range(1, b):
        for c in range(b):
            if (c * (c - 1)) % a == 0:
                if a + 2*c + (c * (c - 1)) // a == b:
                    count += 1
    return count

b = 211
num_beautiful = count_beautiful(b)
print(f"b = {b}, num_beautiful = {num_beautiful}")
```


</turn>

----------------
USER:

<output cell="verify_answer">
b = 211, num_beautiful = 15
</output>

Remaining budget:
 - 84 secs used, 155 secs left,
 - 14974 output tokens used, 1026 output tokens left,
 - 7 thinking steps performed, 3 steps left.
Only 3 left! Make sure you will be ready to answer within the next 3 turns! Adapt your
strategy if necessary.

```
1566    Start your next message by reasoning about how you will solve the task in the next 3 turns.
1567
1568    ----------------
        ASSISTANT:
1569
1570    <turn>
        Okay, I have verified that when $b=211$, the number of $b$-beautiful integers is 15, which
1571    is greater than 10. Since the problem asks for the *least* integer $b$, and I have found a
        $b$ that satisfies the condition, I am confident that the answer is 211.
1572    <return>211</return>
1573
1574    </turn>
```

1575
## Typos, Qwen 2.5 Coder 32B
1576
1577
```
1578    SCORE: 1.0
        Answered in 3 turns and 13 secs with 1367 output tokens (2.36 cts)
1579    COMMENT: CodeAdapt distributes reasoning over language and code by generating pairs of
1580    words with typos and their correction. It leverages code to apply these corrections
        meticulously, but when printing the result, it observes a typo it had missed the first time
1581    and corrects that one before returning the final answer.
1582
1583
1584    --------------------------------
        QUESTION:
1585    Please output this exact text below, with no changes at all except for **fixing the
        misspellings**. Please leave all other stylistic decisions like commas and US vs British
1586    spellings as in the original text. Wrap your final answer between <answer> and </answer>
1587    tags.

1588    Estimating relative camera poses beteen images has been a central probelm in cmoputer
1589    vision. Methods thast fidn correspondences and solve forhten fundamental matrix offer high
        precision in most cases. Conversely, methods predicting pose driectly useing neural
1590    networks are more robust to limited overlap adn can infer absolute translation scale, but
1591    atther expense of reduced precision. We sohw how to combine the best of both methods; our
        approach yields results that are both precice and robust, hwile alos accurately inferring
1592    translation scales. At the heart of our modle lies a Transformer that (1) learns to balance
1593    between solved and learned pose estimations, and (2) provides a prior to guide a solver. A
        comprehensive analysis supports our design choices andd demonstrates that our method adapts
1594    flexibly to various feature extractors anbd correspondence estimators, showing
1595    state-of-teh-art perfomance in 6DoF pose estimation on Matterport3D, InteriorNet,
        StreetLearn, adn Map-free Relocalization.
1596
1597    ----------------
        ASSISTANT:
1598
1599    <turn>
        Let's start by identifying the misspellings in the given text and correcting them. I'll
1600    write a Python function to systematically identify and correct these misspellings.
1601
1602    <code name="correct_misspellings">
        ```python
1603    # Original text with misspellings
        text_with_misspellings = """Estimating relative camera poses beteen images has been a
1604    central probelm in cmoputer vision. Methods thast fidn correspondences and solve forhten
1605    fundamental matrix offer high precision in most cases. Conversely, methods predicting pose
        driectly useing neural networks are more robust to limited overlap adn can infer absolute
1606    translation scale, but atther expense of reduced precision. We sohw how to combine the best
1607    of both methods; our approach yields results that are both precice and robust, hwile alos
        accurately inferring translation scales. At the heart of our modle lies a Transformer that
1608    (1) learns to balance between solved and learned pose estimations, and (2) provides a prior
1609    to guide a solver. A comprehensive analysis supports our design choices andd demonstrates
        that our method adapts flexibly to various feature extractors anbd correspondence
1610    estimators, showing state-of-teh-art perfomance in 6DoF pose estimation on Matterport3D,
1611    InteriorNet, StreetLearn, adn Map-free Relocalization."""

1612    # Dictionary of misspellings and their corrections
1613    misspellings = {
            "beteen": "between",
1614        "probelm": "problem",
            "cmoputer": "computer",
1615        "thast": "that",
            "fidn": "find",
1616        "solve forhten": "solve for the",
            "driectly": "directly",
1617        "useing": "using",
            "adn": "and",
1618        "atther": "at the",
1619
```

```
         "sohw": "show",
         "precice": "precise",
         "hwile": "while",
         "alos": "also",
         "modle": "model",
         "andd": "and",
         "state-of-teh-art": "state-of-the-art",
         "perfomance": "performance"
     }

     # Function to correct misspellings
     def correct_misspellings(text, misspellings):
         for misspelling, correction in misspellings.items():
             text = text.replace(misspelling, correction)
         return text

     # Correcting the text
     corrected_text = correct_misspellings(text_with_misspellings, misspellings)
     print(corrected_text)
     ```

     

     </turn>

     ----------------
     USER:

     <output cell="correct_misspellings">
     Estimating relative camera poses between images has been a central problem in computer
     vision. Methods that find correspondences and solve for the fundamental matrix offer high
     precision in most cases. Conversely, methods predicting pose directly using neural networks
     are more robust to limited overlap and can infer absolute translation scale, but at the
     expense of reduced precision. We show how to combine the best of both methods; our approach
     yields results that are both precise and robust, while also accurately inferring
     translation scales. At the heart of our model lies a Transformer that (1) learns to balance
     between solved and learned pose estimations, and (2) provides a prior to guide a solver. A
     comprehensive analysis supports our design choices and demonstrates that our method adapts
     flexibly to various feature extractors anbd correspondence estimators, showing
     state-of-the-art performance in 6DoF pose estimation on Matterport3D, InteriorNet,
     StreetLearn, and Map-free Relocalization.
     </output>

     Remaining budget:
      - 9 secs used, 230 secs left,
      - 536 output tokens used, 15464 output tokens left,
      - 1 thinking steps performed, 9 steps left.

     ----------------
     ASSISTANT:

     <turn>
     It looks like there is one remaining misspelling in the text: "anbd" should be "and". Let's
     correct that and then return the final corrected text.

     <code name="correct_final_misspelling">
     ```python
     def correct_final_misspelling(text):
         corrections = {
             "anbd": "and"
         }
         for misspelling, correction in corrections.items():
             text = text.replace(misspelling, correction)
         return text

     corrected_text = correct_final_misspelling(corrected_text)
     print(corrected_text)
     ```
     

     </turn>

     ----------------
     USER:

     <output cell="correct_final_misspelling">
     Estimating relative camera poses between images has been a central problem in computer
     vision. Methods that find correspondences and solve for the fundamental matrix offer high
     precision in most cases. Conversely, methods predicting pose directly using neural networks
```

```
1674    are more robust to limited overlap and can infer absolute translation scale, but at the
1675    expense of reduced precision. We show how to combine the best of both methods; our approach
        yields results that are both precise and robust, while also accurately inferring
1676    translation scales. At the heart of our model lies a Transformer that (1) learns to balance
1677    between solved and learned pose estimations, and (2) provides a prior to guide a solver. A
        comprehensive analysis supports our design choices and demonstrates that our method adapts
1678    flexibly to various feature extractors and correspondence estimators, showing
1679    state-of-the-art performance in 6DoF pose estimation on Matterport3D, InteriorNet,
        StreetLearn, and Map-free Relocalization.
1680    </output>
1681
        Remaining budget:
1682     - 11 secs used, 228 secs left,
1683     - 662 output tokens used, 15338 output tokens left,
         - 2 thinking steps performed, 8 steps left.
1684
1685    ----------------
        ASSISTANT:
1686
1687    <turn>
        The final corrected text is ready. Let's return it wrapped between <answer> and </answer>
1688    tags.
1689
1690    <return>
        Estimating relative camera poses between images has been a central problem in computer
1691    vision. Methods that find correspondences and solve for the fundamental matrix offer high
        precision in most cases. Conversely, methods predicting pose directly using neural networks
1692    are more robust to limited overlap and can infer absolute translation scale, but at the
1693    expense of reduced precision. We show how to combine the best of both methods; our approach
        yields results that are both precise and robust, while also accurately inferring
1694    translation scales. At the heart of our model lies a Transformer that (1) learns to balance
1695    between solved and learned pose estimations, and (2) provides a prior to guide a solver. A
        comprehensive analysis supports our design choices and demonstrates that our method adapts
1696    flexibly to various feature extractors and correspondence estimators, showing
1697    state-of-the-art performance in 6DoF pose estimation on Matterport3D, InteriorNet,
        StreetLearn, and Map-free Relocalization.
1698    </return>
1699
1700    </turn>
1701
```

## Zebra, Qwen 3 30B A3B

```
SCORE: 1.0
Answered in 7 turns and 15 secs with 2859 output tokens (3.87 cts)
COMMENT: CodeAct maps this reasoning problem into a constraint-satisfaction problem and
solves it in code.

-------------------------------
QUESTION:
There are 3 people standing in a line. From left to right, they are numbered 1 to 3.
Each person has a set of attributes: Hobby, Movie-Genre, Sport.
The attributes have the following possible values:
Hobby: gardening, rock-climbing, singing
Movie-Genre: drama, fantasy, comedy
Sport: baseball, cricket, water-polo
Each person has a unique value for each attribute.
You know the following about the people:
The person who watches comedy and the person who plays water-polo have different parity
positions
The person who likes rock-climbing is somewhere between the person who watches comedy and
the person who likes singing
The person who watches fantasy is not anywhere to the left of the person who watches comedy
The person who plays baseball is not anywhere to the right of the person who watches fantasy
The person who plays baseball and the person who watches fantasy have different parity
positions
In the above, 'parity position' means the evenness or oddness of the person's position in
the line.
Given this information, answer the following questions:
What is the movie genre of the person who plays cricket?
At what position is the person who watches drama?
At what position is the person who watches fantasy?
What sport does the person who likes singing play?
Think step by step and explain your reasoning, then output your answers in order in the
format:
<solution>answer1, answer2, answer3, ...</solution>
For instance, if there were 3 questions and the answers were A, B, and C, the output would
be:
<solution>A, B, C</solution>
```

```
1728    If the answer to a question is a number, be sure to put it in numerical form (e.g. '3'
1729    instead of 'three').

1730
1731    ----------------
        ASSISTANT:
1732
1733    <turn>
        I need to solve this logic puzzle step by step. There are 3 people in positions 1, 2, 3
1734    (left to right), each with unique values for Hobby, Movie-Genre, and Sport.

1735
        The attributes are:
1736    - Hobby: gardening, rock-climbing, singing
        - Movie-Genre: drama, fantasy, comedy
1737    - Sport: baseball, cricket, water-polo

1738
        I'll represent each person as a dictionary with keys for each attribute, and solve by
1739    systematically applying the constraints.

1740
        Let me first parse all the constraints and convert them into logical conditions:
1741
1742    1. "The person who watches comedy and the person who plays water-polo have different parity
        positions"
1743        comedy position and water-polo position have different parity (one even, one odd)

1744
1745    2. "The person who likes rock-climbing is somewhere between the person who watches comedy
        and the person who likes singing"
1746        rock-climbing is between comedy and singing in position (not necessarily adjacent)

1747
1748    3. "The person who watches fantasy is not anywhere to the left of the person who watches
        comedy"
1749        fantasy position   comedy position

1750
1751    4. "The person who plays baseball is not anywhere to the right of the person who watches
        fantasy"
            baseball position   fantasy position
1752
1753    5. "The person who plays baseball and the person who watches fantasy have different parity
        positions"
1754        baseball position and fantasy position have different parity (one even, one odd)

1755
1756    I'll use a brute-force approach with permutations since 3! = 6 for each attribute, and
        total combinations are 6^3 = 216, which is manageable.

1757
        I'll generate all possible permutations for each attribute and filter based on constraints.
1758
1759    Let me start by writing code to generate all possible configurations and apply constraints.

1760    </turn>

1761
1762    ----------------
        USER:
1763
1764    Your message did not include a code block or return statement.

1765    Please continue problem solving, and remember to only respond with messages that contain
        code blocks or return statements. It might be that you made minor formatting mistakes. If
1766    you are unsure about the format, please refer to the examples.

1767
        Remaining budget:
1768    - 2 secs used, 237 secs left,
        - 369 output tokens used, 15631 output tokens left,
1769    - 1 thinking steps performed, 9 steps left.

1770
1771    ----------------
        ASSISTANT:
1772
1773    <turn>
        <code name="setup">
1774    ```python
        from itertools import permutations
1775
1776    # Define the possible values for each attribute
1777    hobbies = ['gardening', 'rock-climbing', 'singing']
        movie_genres = ['drama', 'fantasy', 'comedy']
1778    sports = ['baseball', 'cricket', 'water-polo']

1779
        # Generate all possible permutations for each attribute
1780    all_hobbies = list(permutations(hobbies))
        all_movie_genres = list(permutations(movie_genres))
1781    all_sports = list(permutations(sports))
```
```

```
1782
1783     # Create a list of all possible configurations
         # Each configuration is a tuple: (hobbies, movie_genres, sports)
1784     # where each is a tuple of length 3 representing positions 1-3
         configurations = [(h, m, s) for h in all_hobbies for m in all_movie_genres for s in
1785     all_sports]
         print(f"Total configurations to check: {len(configurations)}")
1786     ```
1787     
1788
1789     </turn>
1790     ----------------
         USER:
1791
1792     <output cell="setup">
         Total configurations to check: 216
1793     </output>
1794
         Remaining budget:
1795      - 3 secs used, 236 secs left,
1796      - 577 output tokens used, 15423 output tokens left,
         - 2 thinking steps performed, 8 steps left.
1797
1798     ----------------
         ASSISTANT:
1799
1800     <turn>
         Now I'll implement a function to check all the constraints on a given configuration. I'll
1801     go through each constraint one by one and filter out invalid configurations.
1802
1803     Let me define the constraint checking function:
1804
1805
         </turn>
1806
1807     ----------------
         USER:
1808
1809     Your message did not include a code block or return statement.
1810     Please continue problem solving, and remember to only respond with messages that contain
         code blocks or return statements. It might be that you made minor formatting mistakes. If
1811     you are unsure about the format, please refer to the examples.
1812
1813     Remaining budget:
          - 4 secs used, 235 secs left,
1814      - 620 output tokens used, 15380 output tokens left,
         - 3 thinking steps performed, 7 steps left.
1815
1816     ----------------
         ASSISTANT:
1817
1818     <turn>
         <code name="check_constraints">
1819     ```python
         def check_constraints(config):
1820         h, m, s = config # hobbies, movie_genres, sports
1821
             # Find positions (1-indexed) of each value
1822         pos_hobby = {h[i]: i+1 for i in range(3)}
             pos_movie = {m[i]: i+1 for i in range(3)}
1823         pos_sport = {s[i]: i+1 for i in range(3)}
1824
             # Constraint 1: comedy and water-polo have different parity positions
1825         comedy_pos = pos_movie['comedy']
             water_polo_pos = pos_sport['water-polo']
1826         # Different parity means one is odd, one is even
             if (comedy_pos % 2) == (water_polo_pos % 2):
1827             return False
1828
             # Constraint 2: rock-climbing is between comedy and singing
1829         rock_climbing_pos = pos_hobby['rock-climbing']
             comedy_pos = pos_movie['comedy']
1830         singing_pos = pos_hobby['singing']
             # rock-climbing is between comedy and singing if:
1831         # (comedy < rock_climbing < singing) or (singing < rock_climbing < comedy)
             if not ((comedy_pos < rock_climbing_pos < singing_pos) or (singing_pos <
1832     rock_climbing_pos < comedy_pos)):
                 return False
1833
1834
1835
```

```
     # Constraint 3: fantasy is not to the left of comedy
     # fantasy position >= comedy position
     fantasy_pos = pos_movie['fantasy']
     if fantasy_pos < comedy_pos:
         return False

     # Constraint 4: baseball is not to the right of fantasy
     # baseball position <= fantasy position
     baseball_pos = pos_sport['baseball']
     if baseball_pos > fantasy_pos:
         return False

     # Constraint 5: baseball and fantasy have different parity positions
     if (baseball_pos % 2) == (fantasy_pos % 2):
         return False

     return True

# Test the constraint function with a sample configuration
test_config = all_hobbies[0], all_movie_genres[0], all_sports[0]
print(f"Test configuration passes constraints: {check_constraints(test_config)}")
```

```

</turn>

----------------
USER:

<output cell="check_constraints">
Test configuration passes constraints: False
</output>

Remaining budget:
 - 7 secs used, 232 secs left,
 - 1115 output tokens used, 14885 output tokens left,
 - 4 thinking steps performed, 6 steps left.

----------------
ASSISTANT:

<turn>
Now I'll filter all configurations to find the valid ones that satisfy all constraints.
Since there might be multiple valid configurations, I'll check if there's a unique solution
for the required answers.

Let me run the filtering process:

</turn>

----------------
USER:

Your message did not include a code block or return statement.

Please continue problem solving, and remember to only respond with messages that contain
code blocks or return statements. It might be that you made minor formatting mistakes. If
you are unsure about the format, please refer to the examples.

Remaining budget:
 - 8 secs used, 231 secs left,
 - 1164 output tokens used, 14836 output tokens left,
 - 5 thinking steps performed, 5 steps left.

----------------
ASSISTANT:

<turn>
<code name="filter_configurations">
```python
# Filter configurations that satisfy all constraints
valid_configs = []
for config in configurations:
    if check_constraints(config):
        valid_configs.append(config)
```

```
1890    print(f"Number of valid configurations: {len(valid_configs)}")

1891
        # If there's exactly one valid configuration, we can extract the answers
1892    if len(valid_configs) == 1:
            h, m, s = valid_configs[0]
1893        # Extract positions
            pos_hobby = {h[i]: i+1 for i in range(3)}
1894        pos_movie = {m[i]: i+1 for i in range(3)}
            pos_sport = {s[i]: i+1 for i in range(3)}
1895
1896
            # Answer the questions
1897        # 1. What is the movie genre of the person who plays cricket?
            cricket_pos = pos_sport['cricket']
1898        movie_genre_cricket = m[cricket_pos - 1] # m is 0-indexed
1899
            # 2. At what position is the person who watches drama?
1900        drama_pos = pos_movie['drama']
1901
            # 3. At what position is the person who watches fantasy?
1902        fantasy_pos = pos_movie['fantasy']
1903
            # 4. What sport does the person who likes singing play?
1904        singing_pos = pos_hobby['singing']
1905        sport_singing = s[singing_pos - 1] # s is 0-indexed
1906
            print(f"Answer 1 (movie genre of cricket player): {movie_genre_cricket}")
1907        print(f"Answer 2 (position of drama watcher): {drama_pos}")
            print(f"Answer 3 (position of fantasy watcher): {fantasy_pos}")
1908        print(f"Answer 4 (sport of singing person): {sport_singing}")
1909
            # Return the answers in the required format
1910        print(f"<solution>{movie_genre_cricket}, {drama_pos}, {fantasy_pos},
1911    {sport_singing}</solution>")
1912    else:
            print(f"Multiple valid configurations found: {len(valid_configs)}")
1913        # Let's examine the answers across all valid configs
        answers = []
1914    for config in valid_configs:
            h, m, s = config
1915        pos_hobby = {h[i]: i+1 for i in range(3)}
            pos_movie = {m[i]: i+1 for i in range(3)}
1916        pos_sport = {s[i]: i+1 for i in range(3)}
1917
            # Answer 1: movie genre of cricket player
1918        cricket_pos = pos_sport['cricket']
1919        movie_genre_cricket = m[cricket_pos - 1]
1920
            # Answer 2: position of drama watcher
1921        drama_pos = pos_movie['drama']
1922
            # Answer 3: position of fantasy watcher
1923        fantasy_pos = pos_movie['fantasy']
1924
            # Answer 4: sport of singing person
1925        singing_pos = pos_hobby['singing']
1926        sport_singing = s[singing_pos - 1]
1927
            answers.append((movie_genre_cricket, drama_pos, fantasy_pos, sport_singing))
1928
        # Check if all valid configs have the same answers
1929    unique_answers = set(answers)
1930    if len(unique_answers) == 1:
            movie_genre_cricket, drama_pos, fantasy_pos, sport_singing = unique_answers.pop()
1931        print(f"All valid configurations have the same answers:")
            print(f"Answer 1: {movie_genre_cricket}")
1932        print(f"Answer 2: {drama_pos}")
            print(f"Answer 3: {fantasy_pos}")
1933        print(f"Answer 4: {sport_singing}")
            print(f"<solution>{movie_genre_cricket}, {drama_pos}, {fantasy_pos},
1934    {sport_singing}</solution>")
1935    else:
            print(f"Answers vary across valid configurations: {unique_answers}")
1936        # Print all valid configurations for debugging
        for i, config in enumerate(valid_configs):
1937            h, m, s = config
            pos_hobby = {h[i]: i+1 for i in range(3)}
1938        pos_movie = {m[i]: i+1 for i in range(3)}
            pos_sport = {s[i]: i+1 for i in range(3)}
1939
            print(f"Valid config {i+1}:")
1940        print(f"  Hobbies: {h}")
```

```
1944              print(f" Movie-Genres: {m}")
1945              print(f" Sports: {s}")
1946              print(f" Cricket player's movie genre: {m[pos_sport['cricket']-1]}")
1947              print(f" Drama watcher position: {pos_movie['drama']}")
1948              print(f" Fantasy watcher position: {pos_movie['fantasy']}")
1949              print(f" Singing person's sport: {s[pos_hobby['singing']-1]}")
              print()
```



</turn>

----------------
USER:

<output cell="filter_configurations">
Number of valid configurations: 1
Answer 1 (movie genre of cricket player): drama
Answer 2 (position of drama watcher): 3
Answer 3 (position of fantasy watcher): 2
Answer 4 (sport of singing person): cricket
<solution>drama, 3, 2, cricket</solution>
</output>

Remaining budget:
 - 13 secs used, 226 secs left,
 - 2215 output tokens used, 13785 output tokens left,
 - 6 thinking steps performed, 4 steps left.

----------------
ASSISTANT:

<turn>
I have successfully filtered all possible configurations and found exactly one valid
configuration that satisfies all 5 constraints.

The answers are:
1. The movie genre of the person who plays cricket is drama
2. The person who watches drama is at position 3
3. The person who watches fantasy is at position 2
4. The person who likes singing plays cricket

I've verified that this is the only valid configuration, so the answers are unique.

<return><solution>drama, 3, 2, cricket</solution></return>

</turn>

## Memory error due to combinatorial search, Collie, Gemini 2.0 Flash

CONTEXT: In this example, the agent attempts to generate all possible permutations of
letters to create a 13-letter word under constraints. It hits a memory error (max 100MB).

[...]

----------------
ASSISTANT:

<turn>
I see, the issue is that my template replacement is not working as expected. Let me correct
the code.

<code name="generate_combinations_2">
```python
import itertools

def generate_combinations():
    letters = "abcdefghijklmnopqrstuvwxyz"
    for combo in itertools.product(letters, repeat=3):
        yield "".join(combo)

template = list("___s___f__l__")

possible_words = []
for c1, c2, c3, c4, c5, c6, c7, c8, c9, c10 in
itertools.product("abcdefghijklmnopqrstuvwxyz", repeat=10):
```

```
template_copy = template[:]
template_copy[0] = c1
template_copy[1] = c2
template_copy[2] = c3
template_copy[5] = c4
template_copy[6] = c5
template_copy[9] = c6
template_copy[10] = c7
template_copy[12] = c8
word = "".join(template_copy)
if len(word) == 13:
    possible_words.append(word)

print(len(possible_words))
#print(possible_words[:10])
```


<execute name="generate_combinations_2">

</turn>

----------------
USER:

<output cell="generate_combinations_2">

[Execution interrupted due to resource limits]
</output>

<error cell="generate_combinations_2">
Memory limit exceeded: usage grew beyond 500MB
</error>

Remaining budget:
 - 14 secs used, 225 secs left,
 - 1226 output tokens used, 14774 output tokens left,
 - 5 thinking steps performed, 5 steps left.

----------------
ASSISTANT:

<turn>
That's not good. Generating all possible combinations is too memory-intensive. I need a
more efficient approach. I'll try generating a smaller set of combinations and then using
the LLM to refine them.

Given the limited resources, I'll simplify the process. I'll ask the LLM to generate a
single candidate word, and then I'll verify its validity. If it's invalid, I'll have to
return "No valid word found."

[...]
```

## F LANGUAGE MODEL USAGE

Language models are used as a general-purpose assist tool (LM-empowered IDEs, writing refinements, etc.) for the present study. They do not play a significant role in research ideation or paper drafting.