# OpenReview forum: "Code-enabled language models can outperform reasoning models on diverse tasks"
_ICLR.cc/2026/Conference — Submitted to ICLR 2026_

### Official Review · Reviewer_QqoW · 2025-10-25

**Soundness:** 3
**Presentation:** 3
**Contribution:** 1
**Rating:** 2
**Confidence:** 5

**Summary:**

The paper argues that integrating executable code into LLMs fundamentally enhances their reasoning ability. The authors posit that reasoning tasks often require structured computation, which pure text-based models approximate but do not execute. By allowing LLMs to generate, manage, and interpret code within reasoning processes, models can perform verifiable intermediate computations and systematically improve task accuracy. The study evaluates this approach across mathematical and logical reasoning benchmarks, comparing code-enabled models to standard CoT prompting. Results show significant gains in reasoning accuracy and consistency, suggesting that access to a code interpreter reduces hallucination, enforces step validity, and better aligns outputs with formal reasoning logic. There are a number of well-established works along the lines proposed by the authors that use code-based approaches (known as PAL) to solve mathematical, multimodal, and multilingual tasks. The authors have not cited them, and I strongly recommend checking them out.

**Strengths:**

- empirical reuslts: Comparative experiments demonstrate consistent improvements over standard CoT baselines, confirming the practical benefit of code-enabled reasoning.

- clarity: The authors articulate a coherent argument that reasoning can be viewed as “neural-programmatic execution”, bridging formal computation and language understanding.

- impact: The approach is broadly applicable to domains requiring structured reasoning, such as mathematics, data analysis, and theorem proving.

**Weaknesses:**

- Severe figure readability issues: Many figures are illegible, overcrowded, or rendered in fonts too small to interpret. Axes and legends are poorly visible, making it nearly impossible to verify the numerical trends the authors describe. This critically undermines the transparency of their results and the paper’s overall readability. The authors should completely redesign the figures, enlarging fonts, clarifying contrasts, and simplifying layouts to foreground the main findings.

- theoretical grounding: The work is mainly empirical and descriptive, offering little formal analysis of why or when code execution leads to better reasoning.

- Unclear evaluation protocols: The paper does not sufficiently detail the criteria for “successful reasoning” or how correctness is verified when code is executed internally.

- confounds: Improvements may arise from tool-use effects (e.g., calling external functions) rather than intrinsic reasoning advances.

**Questions:**

How do you ensure that improvements from code execution reflect genuine reasoning enhancement rather than simple access to an external computational oracle?

How generalisable is your framework to reasoning tasks that are non-programmatic or linguistically abstract, where code execution provides no direct computational benefit?

---

> ### Author Response · Authors · 2025-11-26
>
> Thank you to the reviewer for your review. We are glad that you recognize that our paper has strong empirical results, is coherently written, and has broad impact and applications.
>
> However, we would like to first point out certain *inaccurate statements* or *factual mistakes* contained in the review.
>
> > The study evaluates this approach across mathematical and logical reasoning benchmarks, comparing code-enabled models to standard CoT prompting.
>
> A key design of our study is that we conduct evaluations on a broad range of tasks beyond just mathematical and logical reasoning, and we compare CodeAdapt to RL-trained reasoning models beyond standard CoT prompting.
>
> > There are a number of well-established works along the lines proposed by the authors that use code-based approaches (known as PAL) to solve mathematical, multimodal, and multilingual tasks. The authors have not cited them, and I strongly recommend checking them out.
>
> We are afraid that this claim is false—we have cited PAL, along with many other important code-based approaches (e.g., “Program of Thought” and CodeAct itself). PAL is cited on Line 75 in the introduction. We believe that we have done a reasonable job at literature review and positioning prior work.
>
> > Severe figure readability issues: Many figures are illegible, overcrowded, or rendered in fonts too small to interpret.
>
> We disagree with this comment. We think our figures are fairly legible by AI/ML paper standards. The main figure has a few sub-figures, and we allocate full paper width to it. One can also find more details of each sub-figure in other parts of the paper: the full trace of (a) is in the appendix, (b) is explained in Alg. 1, and (c) is presented in the main result table. We think the other figures in the paper are even easier to read. Could you let us know which figure you would suggest we “completely redesign”?
>
> > The work is mainly empirical and descriptive, offering little formal analysis of why or when code execution leads to better reasoning.
>
> We disagree. Firstly our paper proposes a framework and algorithm, so it is not entirely accurate to say that it is “mainly empirical and descriptive”, while we think our comprehensive evaluations constitute one of the strengths of the paper, as Reviewer eWPP identifies. Furthermore, we have conducted studies and analyses on the “why or when” in the paper (also identified as “insightful” by eWPP). We have ablation studies in Sec. 4.3, and we investigate how the code-enabled language models actually solve problems in Sec. 4.5. We also include concrete reasoning traces with commentaries in the appendix. Hybrid language+code reasoning is very flexible such that LMs can apply different strategies for different problems/domains—it can help performance across the board, as our results show.
>
> > The paper does not sufficiently detail the criteria for “successful reasoning” or how correctness is verified when code is executed internally.
>
> We have described our benchmarks and correctness metrics in detail in Sec. 4.1 and Appendix A. Correctness is verified the same way across different methods based on the model’s final output.
>
> > How generalisable is your framework to reasoning tasks that are non-programmatic or linguistically abstract, where code execution provides no direct computational benefit?
>
> Indeed, generalizability to other “non-programmatic or linguistically abstract” tasks is a key feature that we empirically show in this work, as we have multiple tasks of this nature. For example, most would agree that all of our “instruction following” and “language processing” benchmarks are of this nature. We see that Reviewers eWPP and xas3 agree that our evaluations are about “multiple” and “diverse” domains.

---

> ### Author Response · Authors · 2025-11-26
>
> Next, we would like to respond to the remaining concern raised by the reviewer.
>
> > Improvements may arise from tool-use effects (e.g., calling external functions) rather than intrinsic reasoning advances. How do you ensure that improvements from code execution reflect genuine reasoning enhancement rather than simple access to an external computational oracle?
>
> Regarding this question, we have two answers. Firstly, we take the stance that it is the overall system (not just CoT) that does the reasoning—code execution is a form of reasoning, and that helps language models. We think this is a mainstream position, backed by earlier work by PAL as you mentioned, especially in the “compound AI systems” [1] and “neurosymbolic AI” [2] communities. Secondly, simplifying having “access to an external computational oracle” is not enough. Our CodeAdapt outperforms the CodeAct baseline, and our GFL algorithm outperforms the BFL baseline. So in this case the claim in our title is only manifested in CodeAdapt, the proposed framework in this work.
>
> We hope our response clarifies any misunderstanding you might have about our paper. If you find them helpful and satisfactory, would you consider re-evaluation and raising the score?
>
> [1] Zaharia, M., Khattab, O., Chen, L., Davis, J. Q., Miller, H., Potts, C., ... & Ghodsi, A. (2024). The shift from models to compound AI systems. Berkeley Artificial Intelligence Research Lab. Available online at: https://bair. berkeley. edu/blog/2024/02/18/compound-ai-systems/(accessed February 27, 2024).
>
> [2] Bhuyan, B. P., Ramdane-Cherif, A., Tomar, R., & Singh, T. P. (2024). Neuro-symbolic artificial intelligence: a survey. Neural Computing and Applications, 36(21), 12809-12844.

---

### Official Review · Reviewer_xas3 · 2025-11-01

**Soundness:** 2
**Presentation:** 2
**Contribution:** 1
**Rating:** 2
**Confidence:** 4

**Summary:**

This paper proposes CodeAdapt, which incorporates few-shot bootstrap in-context learning based on CodeAct. CodeAdapt achieves performance improvements in areas such as instruction following, language processing, and formal reasoning.

**Strengths:**

1. By adding few-shot bootstrap in-context learning on top of CodeAct, it enables self-exploration of reasoning trajectories, eliminating the need for expert demonstrations.
2. It achieves performance improvements across multiple domains.

**Weaknesses:**

1. The main issue with this paper is the lack of originality. Using code as a form of reasoning has already been widely studied in previous work. This paper mainly adds few-shot in-context learning on top of CodeAct, which does not provide substantial new insights. Domain adaptation through in-context examples has also been extensively explored in prior research.

2. The paper lacks comparisons with other few-shot in-context learning or few-shot domain adaptation methods.

**Questions:**

Please see weaknesses.

---

> ### Author Response · Authors · 2025-11-26
>
> Thank you to the reviewer for your review. We are encouraged that you acknowledge that our method enables self-exploration, does not require expert demonstrations, and achieves performance improvements. Next we address the weaknesses below point by point.
>
> > The main issue with this paper is the lack of originality.
>
> We have responded to this issue in the general response. More specifically here, even though “[d]omain adaptation through in-context examples has also been extensively explored” for reasoning, that happens mostly in the context of chain-of-thought, whereas our focus is code-enhanced reasoning—a different paradigm. For this work, the intended “substantial new” insight is how well code-enhanced multi-step reasoning without finetuning can work when comparing against reasoning models. Given that the performance is achieved by learning from only a few problems, we expect that (1) the findings are unexpected to many in the community and (2) the simplicity of CodeAdapt may directly have an impact on real-world deployment and applications of language model reasoning.
>
> > The paper lacks comparisons with other few-shot in-context learning or few-shot domain adaptation methods.
>
> Regarding this, we have a two-part response. Firstly, we have BFL (bootstrap few-shot learning) as a baseline, which we think is a reasonable choice (especially given that we operate in the context of self-generated trajectories / no expert demonstration). We show that GFL meaningfully outperforms BFL. More importantly, as articulated above, the main contribution of this paper is the scientific finding that with CodeAdapt instruct models in many cases outperform the corresponding reasoning models. CodeAdapt (but not just CodeAct or BFL) suffices to establish the claim. We have stated in the paper (Line 476) that we do not claim that GFL is the SoTA few-shot in-context learning method. If another method M outperforms GFL, that would mean M would allow instruct models to further outperform reasoning models and thus make our position stronger. So we can reasonably leave the search for even better-performing methods for future work, while maintaining that code-enhanced reasoning is a powerful reasoning paradigm that is worthy of more scientific and engineering investigations.

---

### Official Review · Reviewer_eWPP · 2025-11-01

**Soundness:** 3
**Presentation:** 3
**Contribution:** 3
**Rating:** 6
**Confidence:** 5

**Summary:**

This paper presents a compelling study on how code-enabled language models (LMs) can achieve reasoning capabilities comparable to or even surpassing specialized reasoning models (RMs) without expensive reinforcement learning training. By integrating iterative code execution (CodeAct) with lightweight in-context learning (CodeAdapt), the authors demonstrate that ordinary LMs can excel across diverse tasks—from instruction following to mathematical reasoning—using only a handful of training examples. The work highlights a cost-effective and efficient alternative to resource-intensive RM training, contributing significantly to the advancement of hybrid reasoning systems in AI.

**Strengths:**

- The research addresses a timely and novel question—whether code-augmented LMs can compete with expensively trained RMs—offering a fresh perspective on resource-efficient AI reasoning.
- The proposed CodeAdapt framework is both effective and practical, achieving superior performance across multiple tasks with minimal data and computational overhead, as validated by extensive experiments.
- The paper is well-structured and clearly written, with comprehensive evaluations, ablation studies, and insightful analyses of reasoning patterns and resource usage.

**Weaknesses:**

While the study is thorough, future work could explore the scalability of CodeAdapt to a broader range of models and real-world applications to further strengthen its generalizability.

**Questions:**

- Using code to enhance language models is very common in current LLM research, which is why I'm concerned about the limited novelty of this paper. Can you summarize your core innovations again?
- In the experiments, is the baseline design too simple?

---

> ### Author Response · Authors · 2025-11-26
>
> We thank the reviewer for your thoughtful comments. We appreciate your positive assessment—finding the study timely, the framework effective and practical, and the paper clearly written. We agree on the point about future work. We think CodeAdapt opens up new avenues for deeper exploration and understanding of code-enhanced multi-step reasoning, and studying scalability and real-world applications are great directions.
>
> We have addressed the point about novelty in the general response above. Here’s our comment on your question
>
> > In the experiments, is the baseline design too simple?
>
> Thanks for considering this. In our opinion, our choice of baselines suffice for supporting our main claims. In addition to the reasoning models, we compare against CoT prompting, CodeAct, and a simpler in-context learning algorithm (BFL). Together with the ablation studies, we show that both multi-step CodeAct (reasoning format) and GFL (learning algorithm) are needed for significantly improved performance that match or surpass reasoning models. If a “baseline” exists that performs even better than CodeAdapt, that would actually make the central points of the paper stronger (hybrid reasoning is powerful and may represent an important paradigm beyond CoT for both prompt optimization and in-weight RL).
>
> We hope our response clarifies your questions and please let us know if you have additional comments or would consider raising the score.

---

### Official Review · Reviewer_2f2z · 2025-11-01

**Soundness:** 2
**Presentation:** 3
**Contribution:** 2
**Rating:** 2
**Confidence:** 4

**Summary:**

This paper introduces CodeAdapt, a framework that equips LLMs with the ability to perform complex reasoning by interleaving natural language with code execution, combined with a lightweight, few-shot in-context learning procedure. The central claim is that this approach allows non-reasoning LLMs to match or even surpass the performance of specialized, expensively trained reasoning models across a diverse set of tasks.

**Strengths:**

1. Enhancing the reasoning capabilities of models under low-resource conditions is a topic worthy of research.
2. The paper is easy to read.

**Weaknesses:**

1. The contribution is limited, which is a small incremental work upon CodeAct.
2. The work does not compare with other training-free and training-based methods for enhancing LLM reasoning.
3. The experimental setup is limited to 30/32B and API-based LLMs. The lack of experiments with 7B models raises questions about whether the method's effectiveness is overly dependent on the inherent reasoning capabilities of the LLMs.
4. The dataset used in the paper is relatively small, which is insufficient to verify the robustness of the proposed method.
5. What is the performance of the reasoning LLMs with general in-context learning? The proposed method uses 2-shot for in-context learning, making the comparison not fair.

**Questions:**

Please see the weaknesses.

---

> ### Author Response · Authors · 2025-11-26
>
> We thank the reviewer for your thoughtful comments and feedback. We appreciate that you find the topic important and the paper easy to read. Here we address the concerns below point by point.
>
> > The contribution is limited, which is a small incremental work upon CodeAct.
>
> We have responded to this point in the general response above.
>
> > The work does not compare with other training-free and training-based methods for enhancing LLM reasoning.
>
> Firstly, we compare CodeAdapt with CoT, CodeAct, and a simpler in-context bootstrapping method (BFL), in addition to strong reasoning models. The comparisons indicate that CodeAdapt robustly balances performance and efficiency. Secondly, we think our experiments are thorough enough to establish our findings (as Reviewer eWPP points out). The point of the paper is not necessarily that CodeAdapt is the SoTA compared to all methods (as stated in Sec. 5), but we lay out a substantial claim with comprehensive experiments to support. The choices of baselines and ablation studies are for serving that purpose.
>
> > The experimental setup is limited to 30/32B and API-based LLMs.
>
> While we have acknowledged that model coverage is a limitation in Sec. 5 of the main text, we think our model choice—including both open and close models and different generations of Qwen models—is sensible and has variations. Note that in our design each pair of instruct and reasoning models match in size, so under charitable interpretation our results are not “overly dependent on the inherent reasoning capabilities of the LLMs”, because the reasoning capabilities of the corresponding reasoning models are stronger. When it comes to size, our intuition is that DeepSeek V3 outperforming R1 (the biggest pair of models studied here) is the most surprising, since R1 is such a capable model. We are the first to show that one can bring V3 up to R1-level performance without reinforcement learning, making the work more unique.
>
> > The dataset used in the paper is relatively small, which is insufficient to verify the robustness of the proposed method.
>
> We agree that benchmark size can be a general limitation, but we think this is not a weakness that particularly applies to our work for a few reasons. (1) For most benchmarks we take them directly from existing published work (e.g., LiveBench and MuSR). (2) Benchmarks like IFBench and Collie are not small. (3) We have actively made design choices to mitigate small test sets. Many papers treat AIME 2024 and 2025 as separate benchmarks while each only has 30 problems. We combine problems from 2023 to 2025 as one benchmark, intending to establish more robustness.
>
> Importantly, we have run proper statistical significance tests for the main results, and our findings are statistically significant (see Table 1). Thus technically speaking the size of the benchmarks should not be viewed as an issue.
>
> > What is the performance of the reasoning LLMs with general in-context learning? The proposed method uses 2-shot for in-context learning, making the comparison not fair.
>
> Regarding this, we think it is fair to use 2-shot for in-context learning and compare it to reasoning models for two reasons. The first is about the argument structure of this work. Let us use > to denote the relation “performs better than”. We know "prompt optimization < reinforcement learning" for reasoning. Here we show "prompt optimization + code >= reinforcement learning". This makes sense because RL is expensive while prompt opt. is much cheaper (including in the code setting). Thus whether RL + prompt opt. is better than CodeAdapt does not affect our contributions. The second reason is about real-world typical use cases. DeepSeek explicitly states that “When evaluating DeepSeek-R1, we observe that it is sensitive to prompts. Few-shot prompting consistently degrades its performance. Therefore, we recommend users directly describe the problem and specify the output format using a zero-shot setting for optimal results” (arXiv v1, pp.16). And OpenAI has made similar suggestions when o1 was released. As a result, reasoning models are typically used in 0-shot settings, whereas people employ prompting strategies for instruct models. Thus our evaluations reflect real-world situations.
>
> Please let us know if our responses address your concerns and consider raising the score if so.

---

### Author Response · Authors · 2025-11-26
**General response**

We thank the reviewers for their engagement and feedback. We appreciate that the reviewers find that:
1. **enhancing language model reasoning** with **resource efficiency** (in our case sample, token, and time efficiency) is **timely** and **worthy** of research (2f2z, eWPP),
2. the paper is **clearly written** (2f2z, eWPP, QqoW), and
3. our proposed CodeAdapt framework is **effective, practical, and broadly applicable** without the need for expert demonstrations (eWPP, xas3, QqoW).

The main point raised by several reviewers is the novelty/originality of this work, and we would like to address this point here. We have addressed all other concerns in the individual responses.

- An important part of our novel contributions is the **scientific findings** themselves, which many would find surprising or unexpected. The community knows that code can improve reasoning, but previously we do not know how much it can help with respect to reasoning models, and how general or diverse code-enhanced hybrid reasoning can be applied to. Our results are encouraging and positive, and our claim that code-enhanced hybrid reasoning enables instruct models to outperform reasoning models on a wide range of domains is solidly supported by the well designed and executed experiments.
- To our knowledge our paper is the **first** that substantially argues and shows that the multi-step hybrid language+code format is beneficial for general-purpose reasoning (e.g., including natural language generation tasks, beyond math/logic/STEM), as evident from the reasoning model comparisons. One can see this by contrasting our work with those discussed in [1], a recently published survey paper.
- CodeAdapt does not only allow instruct models to perform better than reasoning models—it is also more resource efficient in terms of both **tokens** (up to 81%) and **time** (up to 48 %) for all models (Table 3, Section 4.4, and Appendix D). To our knowledge our work is also first to thoroughly demonstrate the resource advantage of multi-step hybrid reasoning in such settings, which makes this kind of reasoning even more attractive and worthy of further investigations. We believe that this alone is a scientific contribution that is of broad interest.
- We think the fact that such performance gains are achieved under **5 training problems** per domain without any human expert demonstration or finetuning is non-obvious for the reader and constitutes valuable knowledge for the community. Our approach is completely domain general in the sense that no specific scaffolding is needed. And this is achieved by our intuitive yet **novel** few-shot learning algorithm.
- Our findings can have significant **implications**. For one, because of the simplicity and effectiveness of our framework, one can argue that it is production ready for incorporating into real-world applications. For another, as we articulate in both the abstract and the main text, our work suggests that it is promising to do RL on top of the code-enhanced hybrid format for general purpose reasoning. This may have downstream implications for standard model training pipelines.
- Lastly, we view the code-enhanced hybrid reasoning format as a different/complementary paradigm with respect to language-only CoT. A large number of papers has been published in the past three years on evaluating, analyzing and improving CoT reasoning. If it is possible for code-enhanced hybrid reasoning to have a similar status, then it is not fair to say that our work is not novel only because CodeAct exists. Not only is our work valuable and convincing, our results suggest we should want more work to study this kind of reasoning to gain **deeper understanding** that may unlock further progress.

[1] Yang, D., Liu, T., Zhang, D., Simoulin, A., Liu, X., Cao, Y., ... & McAuley, J. (2025). Code to think, think to code: A survey on code-enhanced reasoning and reasoning-driven code intelligence in llms. arXiv preprint arXiv:2502.19411.

---

### Meta-Review · Area_Chair_2iQp · 2026-01-07

**Summary:**

This work presents a compelling study on how code-enabled language models (LMs) can obtain reasoning capabilities comparable to or even surpassing specialized reasoning models (RMs) without expensive reinforcement learning training. By adding few-shot bootstrap in-context learning on top of CodeAct, the approach enables self-exploration of reasoning trajectories, eliminating the need for expert demonstrations. The method achieves performance improvements across multiple domains. The technical contributions are limited, which are small incremental work upon CodeAct. The dataset used in the paper is relatively small, which is not sufficient enough to verify the robustness of the proposed approach. Most of the reviewers recommend rejection.

**Reviewer Concerns:**

The technical contributions are limited. The dataset used in the paper is relatively small, which is not sufficient enough to verify the robustness of the proposed approach. These concerns are not well addressed.

**Reviewer Scores:**

Reviewers are likely to keep the rating.

---

### Decision · Program_Chairs · 2026-01-26

Reject